# SCAFFOLD-AWARE GENERATIVE AUGMENTATION AND RERANKING FOR ENHANCED VIRTUAL SCREENING

## ABSTRACT

Ligand-based virtual screening (VS) is an essential step in drug discovery that evaluates large chemical libraries to identify compounds that potentially bind to a therapeutic target. However, VS faces three major challenges: class imbalance due to the low active rate, structural imbalance among active molecules where certain scaffolds dominate, and the need to identify structurally diverse active compounds for novel drug development. We introduce **ScaffAug**, a scaffold-aware VS framework that addresses these challenges through three modules. The *augmentation module* first generates synthetic data conditioned on scaffolds of actual hits using generative models, specifically a graph diffusion model. This helps mitigate the class imbalance and furthermore the structural imbalance, due to our proposed scaffold-aware sampling algorithm, designed to produce more samples for active molecules with underrepresented scaffolds. A model-agnostic *self-training module* is then used to safely integrate the generated synthetic data from our augmentation module with the original labeled data. Lastly, we introduce a *reranking module* that improves VS by enhancing scaffold diversity in the top recommended set of molecules, while still maintaining and even enhancing the overall general performance of identifying novel, active compounds. We conduct comprehensive computational experiments across five target classes, comparing ScaffAug against existing baseline methods by reporting the performance of multiple evaluation metrics and performing ablation studies on ScaffAug. Overall, this work introduces novel perspectives on effectively enhancing VS by leveraging generative augmentations, reranking, and general scaffold-awareness.

## 1 INTRODUCTION

Virtual screening (VS) supports traditional drug discovery through computational identification of promising compounds for experimental validation, reducing time, cost, and labor (Leelananda & Lindert, 2016). A fundamental task in VS involves predicting the biological activity of chemical compounds against specific therapeutic targets based on their molecular structures. Recent advances in deep learning methods, particularly graph neural networks (GNNs), have significantly improved the effectiveness of these predictive tasks (Golkov et al., 2020). However, three key challenges remain. The first challenge is **class imbalance**, which occurs because large libraries of molecules are tested in high-throughput screening (HTS), but only a small percentage are actual actives (Zhu et al., 2013). This means limited known, active molecules are available for reference, making model training difficult and biasing predictions toward the far more numerous inactive molecules. The second challenge is **structural imbalance** among known active molecules in the training data. This imbalance among known active molecules is commonly observed in VS datasets like WelQrate (Liu et al., 2024b), where some groups of similar active molecules form dominant clusters. Meanwhile, many other active molecules possess structures significantly different from those in the clusters, making them underrepresented in the data. As a result, models become biased toward dominant structural clusters, potentially overlooking novel candidates with distinct molecular structures. The third challenge arises from the requirement in actual drug discovery applications to **discover novel molecules** distinct from existing patented structures. To increase the probability of finding novel active compounds across different regions of chemical space, medicinal chemists maximize scaffold diversity when constructing compound libraries for HTS (Langdon et al., 2011) and creating virtual screening libraries (Krier et al., 2006). This involves clustering molecules based on their core

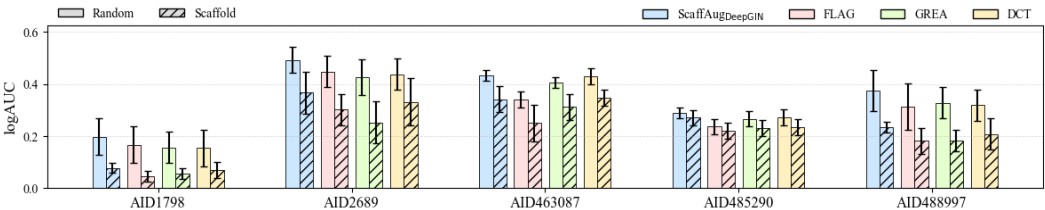

Figure 1: Representative overview of ScaffAug's performance compared to established baseline methods on benchmark datasets. The figure highlights improvements achieved through our scaffold-aware augmentation and reranking framework. Comprehensive experimental analyses, including additional metrics and ablation studies, are presented in Section 4.

structures (scaffolds) and selecting representative compounds from each cluster, with protocols like DEKOIS 2.0 (Bauer et al., 2013), which selects the most potent compound from distinct structural clusters. Deep learning models often struggle to identify novel active molecules because they tend to overfit and prioritize compounds with structural features similar to active molecules in the training data, rather than recognizing diverse structural patterns that may still produce biological activity.

We propose to leverage synthetic data generation (Lu et al., 2023) through generative models, specifically a graph diffusion model (Vignac et al., 2022) for small molecules to address both class and structural imbalance challenges. This method creates new samples conditioned on known active molecules' scaffolds to generate likely active compounds, effectively balancing the representation of active versus inactive molecules in the training data. By directing the generation process toward underrepresented molecular scaffolds via our proposed scaffold-aware sampling, we produce sufficient samples that ensure adequate representation of minor structural families, enabling models to learn more comprehensive structure-activity relationships beyond dominant structural clusters. We note that this approach overcomes the inherent challenges of directly generating molecules and instead integrates the benefits of molecular generation into our enhanced virtual screening framework. To tackle the third challenge of discovering novel molecules, we design a reranking algorithm that injects scaffold diversity consideration into deep learning model's top prediction set output. This approach helps identify promising candidates with structural novelty that models' predicted activity scores might otherwise overlook.

Specifically, we introduce **ScaffAug**, a novel scaffold-aware generative augmentation and reranking framework that consists of three modules to tackle the above three challenges. We demonstrate that ScaffAug significantly outperforms prior baselines across diverse VS tasks, as shown in Figure 1.

- In the augmentation module, we employ two key steps: a scaffold-aware sampling(SAS) strategy to address structural imbalances among known active molecules and build a scaffold library, and scaffold extension using a graph diffusion model (GDM) (Vignac et al., 2022) that generates novel molecules while preserving the core scaffold structures in the library. The resulting generative diverse scaffold-augmented(G-DSA) dataset is then used for the training module.

- The model-agnostic self-training module safely combines the augmentation dataset with the original labeled data with a pseudo-labeling strategy. An illustrative comparison with baselines is shown in Fig. 1, while the full experimental results are presented in Section 4.

- In the reranking module, we apply Maximal Marginal Relevance (MMR) (Carbonell & Goldstein, 1998) to encourage scaffold diversity in the model's top predicted set. This approach balances the model's predicted scores with a diversity score, reranking the list to achieve higher scaffold diversity while maintaining screening quality.

The rest of the paper is organized as follows. In Section 2, we introduce preliminaries covering problem definition, notations, and related work. Thereafter, we present our proposed ScaffAug framework in Section 3. Our experimental setup, including datasets, baseline methods, and evaluation metrics are presented in Section 4. Then, Section 5 discusses the main experimental results, along with ablation studies and limitations. Lastly, we conclude in Section 6.

## 2 PRELIMINARIES

### 2.1 PROBLEM DEFINITION: LIGAND-BASED VIRTUAL SCREENING

Ligand-based virtual screening(VS) identifies potential drug candidates from a large molecule library. We represent molecules as 2D graphs $\mathcal{G}$, where nodes and edges are defined by categorical attributes in spaces $\mathcal{X}$ and $\mathcal{E}$ with cardinalities $|\mathcal{X}| = a$ and $|\mathcal{E}| = b$. The values $a$ and $b$ correspond to the number of atom types (e.g., carbon) and bond types (e.g., single bond) considered. We denote the attribute of node $i$ by $x_i \in \mathbb{R}^a$, which is the one-hot encoding of its node type. Node features of a molecular graph are arranged in a matrix $\mathbf{X} \in \mathbb{R}^{n \times a}$, where $n$ is the node count. The connectivity between nodes and their edge attributes is represented by a tensor $\mathbf{E} \in \mathbb{R}^{n \times n \times b}$, where the edge attribute $e_{ij} \in \mathbb{R}^b$. VS seeks to solve a binary classification task $f_\phi : \mathcal{G} \to \{0, 1\}$, based on training data where a molecule with label one indicates an active hit to the protein target. In practice, we commonly evaluate VS models' ranking abilities because only top-$k$ molecules with the highest prediction scores are valued and will be further validated by wet-lab experiments by domain experts.

### 2.2 RELATED WORK

**Augmentations on molecular graphs for enhanced virtual screening.** Graph-level augmentation improves training on imbalanced data by adding auxiliary information (Ma et al., 2025). Examples include G-mixup, which interpolates graph generators (graphons) across classes (Han et al., 2022), and G$^2$GNN, which builds a graph-of-graphs to enrich minority class context (Wang et al., 2022). FLAG (Kong et al., 2022) perturbs node features adversarially, while GREA (Liu et al., 2022) augments subgraphs. However, such general-purpose methods ignore chemical valency rules, often producing invalid molecules that hinder training. G$^2$GNN is further limited by scalability, as it requires pairwise similarity across all samples. By contrast, DCT (Liu et al., 2023a), a chemistry-specific diffusion model, generates valid task-specific examples, though its benefits for virtual screening remain uncertain given the extreme class imbalance discussed in Sec. 1. InstructMol (Cao et al., 2025) introduces a novel semi-supervised framework that improves molecular property prediction by using a target model to generate pseudo-labels for unlabeled molecules and an instructor model to evaluate their reliability.

**Generative models for small molecule design.** AI-driven molecule design spans variational autoencoders (Blaschke et al., 2018), BERT-style models (Wang et al., 2019), GANs (De Cao & Kipf, 2018), and graph neural networks (Li et al., 2018). Recent work has adapted GPT-like language models for molecular generation (Bagal et al., 2021; Bagal et al.). Diffusion models, first developed for image synthesis (Ho et al., 2020), are now widely used across domains, including molecular graphs, where they address permutation invariance and discrete structure (Niu et al., 2020; Vignac et al., 2022). Conditional variants guide generation by imposing structural or property constraints (Huang et al., 2023; Liu et al., 2024a). In our work, we employ DiGress (Vignac et al., 2022) to generate molecules conditioned on scaffolds of known actives, thus steering synthesis toward chemically meaningful regions. Integrated into our augmentation framework, this conditional approach produces structurally diverse molecules and helps alleviate both class and structural imbalance in VS datasets.

## 3 SCAFFAUG FRAMEWORK

Here, we introduce the ScaffAug framework, visualized in Figure 2, which addresses the three challenges in VS via three modules. The augmentation generates structurally diverse synthetic samples by extending scaffolds from known actives using a graph diffusion model, addressing the class and structural imbalance challenges. The self-training module incorporates these synthetic molecules into training via confidence-based pseudo-labeling. The reranking module improves the structural diversity in top-ranked compounds while maintaining the hit rate against the target protein.

### 3.1 AUGMENTATION MODULE: MITIGATING CLASS AND SCAFFOLD IMBALANCES

The augmentation module, as shown in Figure 2(a), generates synthetic data to address the class imbalance by increasing the proportion of active molecules. To ensure the synthetic data remains within the active molecule distribution, we preserve the scaffolds of active compounds and generate new molecules with a graph diffusion model, a process we call scaffold extension. Before this step,

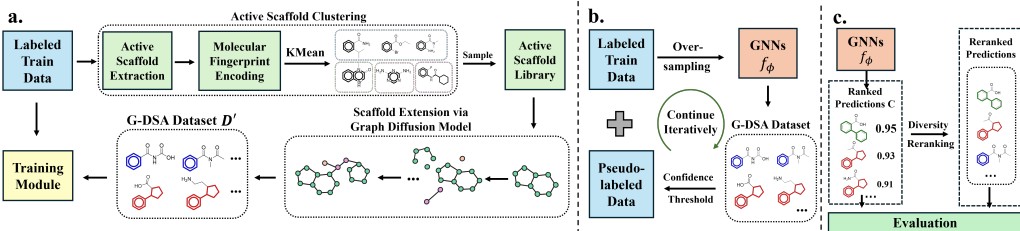

Figure 2: **ScaffAug** Framework. (a) The augmentation module first starts with the scaffold-aware sampling algorithm, which addresses the structural imbalance among active molecules. It clusters active scaffolds based on structure and favors sampling underrepresented scaffolds when constructing the scaffold library. GDM then generates new synthetic samples conditioned on scaffolds in the scaffold library to form the generative diverse scaffold-augmented (G-DSA) dataset. (b) The self-training module incorporates labeled data and augmentation data with a pseudo-labeling strategy. (c) The reranking module leverages prediction scores output by models to produce a more structurally diverse set of top compounds.

we use a scaffold-aware sampling(SAS) algorithm to balance the dominant and underrepresented active scaffolds when building the scaffold library, tackling the structural imbalance shown in Fig. 5. We prioritize underrepresented scaffolds from the known, active molecules for the scaffold extension.

**Scaffold-aware Sampling (SAS) Algorithm**. We begin by extracting scaffold SMILES strings from active molecules in the training set. These scaffolds are then converted to 1024-bit Extended-Connectivity Fingerprints (ECFP)(Rogers & Hahn, 2010) to enable structural similarity measurement. We apply K-Means(MacQueen, 1967) clustering to assign scaffold labels, with the optimal number of clusters determined by Silhouette scores(Rousseeuw, 1987). Our sampling procedure calculates weights inversely proportional to cluster frequencies, giving higher priority to less common structural patterns. After normalizing these weights into sampling probabilities, we select $N$ scaffolds with replacement using these probabilities, where $N$ equals 10% of the training data size. We denote the resulting scaffold graph library as $\mathcal{S} = \{s_j\}_{j=1}^N$, which serves as the foundation for the scaffold extension process. Details of the algorithm are provided in Alg. 1.

---

**Algorithm 1** Scaffold-aware Sampling(SAS)

---

1: **procedure** SCAFFOLDCLUSTERING($\{s_i\}_{i=1}^M$)
2:     **Input:** Active Scaffold SMILES $\{s_i\}$ for $M$ molecules
3:     Convert each $s_i$ to Morgan fingerprint $s_i' \in \mathbb{R}^{1024}$ (radius=2)
4:     Cluster fingerprints with KMeans to get cluster IDs $\{c_i\}$
5:     **return** $\{c_i\}_{i=1}^M$
6: **end procedure**
7: **procedure** SAMPLING($\{(c_i, s_i)\}_{i=1}^M, N$), where $N < M$
8:     **Input:** Cluster assignments $\{c_i\}$, sample size $N$
9:     $N_s[c] \leftarrow \sum_{i=1}^M \mathbf{1}(c_i = c)$            ▷ Compute scaffold cluster counts
10:     $w_s[c] \leftarrow 1/(N_s[c] + \epsilon)$                 ▷ Compute weights
11:     Normalize $w_s$ to probabilities $P_c$ such that $\{P_c\}_{i=1}^M = 1$
12:     Sample $N$ indices $\{i_k\}_{k=1}^N$ with replacement using $\{P_c\}$
13:     **return** Selected scaffolds $\{s_{i_k}\}_{k=1}^N$
14: **end procedure**

---

**Scaffold Extension via Graph Diffusion Model**. After constructing the scaffold library with the SAS algorithm, scaffold extension is performed on those scaffolds to generate synthetic data. We adopt DiGress(Vignac et al., 2022), a discrete diffusion model for graph generation that diffuses and denoises the molecular graph in the discrete state space while maintaining state-of-the-art performance on unconditional molecule graph generation. Our DiGress is trained on 450,000 unlabeled, drug-like molecules, and its sampling algorithm is modified for the scaffold extension task.

Given a scaffold graph $s_i \in \mathcal{S}$ with $n'$ atoms, we sample $n > n'$ from the molecular size distribution of unlabeled data to ensure that we can generate a new molecule larger than the scaffold. During the sampling process of DiGress, we first sample $G^T \sim q_X(n) \times q_E(n)$ where $q_X, q_E$ are prior distributions pre-computed by the marginal distribution of each node and edge type. We define the number of reverse steps $T = 50$, which is sufficient for DiGress to generate valid molecules when fixing the scaffold. Next, we derive a scaffold mask $m$ indicating the atoms and bonds' positions of $s_i$ in $G^T$. We condition the sampling process by anchoring the scaffold substructure $s_i$ and allowing variation in the remaining substructure. During each time step $t$ of the reverse diffusion process, we sample the unknown regions of the molecular graph based on the predictions of the denoising network $p_\theta$ while fixing the scaffold part:

$$G^t = m \odot s + (1 - m) \odot G^t \quad \text{and} \quad G^{t-1} \sim p_\theta(G^{t-1} \mid G^t). \tag{1}$$

The full sampling algorithm is illustrated in the Appendix A.5. We generate a new molecule based on every $s_i \in \mathcal{S}$ and keep the chemically valid ones to build up a generative diverse scaffold-augmented (G-DSA) dataset $\mathcal{D}'$ for the self-training module.

### 3.2 Self-Training Module: Integrating Generated Molecules Safely in VS

The self-training module, as illustrated in Figure 2(b), leverages the G-DSA dataset $\mathcal{D}'$ as an unlabeled resource. The self-training(Amini et al., 2025) process comprises two phases: an initial warm-up period and subsequent pseudo-labeling cycles. During the warm-up phase, which spans the first $E_{start}$ epochs, the activity predictor model $f_\phi$ trains exclusively on the original labeled dataset $\mathcal{D}$ to establish an initial understanding of active molecule space. Following the warm-up, we implement an iterative pseudo-labeling strategy that executes every $E_{freq}$ epochs. At the start of each interval, the model evaluates all samples in the G-DSA dataset $\mathcal{D}'$ and assigns pseudo-labels only to those molecules where prediction confidence exceeds a predefined threshold $\tau$, creating a subset $\mathcal{D}'_{conf} = \{G_i \mid f_\phi(G_i) > \tau, G_i \in \mathcal{D}'\}$ containing only the most reliable pseudo-labeled examples. For subsequent training iterations until the next pseudo-labeling cycle, we optimize $f_\phi$ on the augmented dataset formed by merging the original labeled data with confidently pseudo-labeled samples $\mathcal{D} \cup \mathcal{D}'_{conf}$. Details of the algorithm are provided in Appendix A.6.

### 3.3 Reranking Module: Preserving Scaffold Diversity in the Top-ranked Set

Virtual screening is typically handled as a ranking problem because pharmaceutical companies only purchase or synthesize the top-ranked molecules, while those with lower prediction scores receive less attention. To tackle the challenge of identifying novel active compounds, we design a Maximal Marginal Relevance (MMR) (Carbonell & Goldstein, 1998) reranking algorithm to enhance the structural diversity within the top-ranked set initially output by the VS model $f_\phi$.

**Reranking Algorithm**. The MMR-based reranking, shown in Alg. 2, selects molecules iteratively in the top-predicted list by balancing the model's prediction scores and structural diversity. Given a top-$k$ predicted set $\mathcal{C} = \{(c_i, p_i)\}_{i=1}^k$ where $p_i = f_\phi(c_i)$, the algorithm first removes the highest-scoring molecule from $\mathcal{C}$ and adds it to the reranked set $\mathcal{R}$. Then, for the remaining iterations, it evaluates every candidate molecule $c_i \in \mathcal{C}$ by calculating an MMR score, which combines $p_i$ with the maximum similarity between $c_i$ and any molecule $r_j \in \mathcal{R}$ already in the reranked set. The molecule with the highest MMR score is selected in each iteration. The algorithm ends when the last molecule is moved from the candidate set to the reranked set.

## 4 Experiments

### 4.1 Experimental Setup

**Datasets**. For this work, we utilize the comprehensive WelQrate (Liu et al., 2024b) dataset, which is a high-quality collection of nine bioassays covering five therapeutic target classes for small-molecule drug discovery benchmarking. Its quality stems from a rigorous hierarchical curation process that includes primary screens followed by confirmatory and counter screens, along with extensive filtering(e.g., PAINS filtering, druglikeness assessment) and expert verification. The datasets realistically represent drug discovery campaigns with large compound numbers ($\sim$66K-300K) and low

---

**Algorithm 2** Molecule Reranking via MMR

---

**Require:** Top-$k$ predicted set $\mathcal{C}$, similarity function $\text{Sim}(c_i, c_j)$, trade-off parameter $\lambda$
**Ensure:** Reranked set $\mathcal{R}$
1: Initialize $\mathcal{R} \leftarrow \emptyset$
2: Select $c^* \leftarrow \arg\max_{c_i \in \mathcal{C}} p_i$              $\triangleright$ Start with highest scoring molecule
3: $\mathcal{R} \leftarrow \mathcal{R} \cup \{c^*\}, \mathcal{C} \leftarrow \mathcal{C} \setminus \{c^*\}$
4: **while** $\mathcal{C} \neq \emptyset$ **do**
5:     **for** each molecule $c_i \in \mathcal{C}$ **do**
6:         $\max \text{Sim}_i \leftarrow \max_{r_j \in \mathcal{R}} \text{Sim}(c_i, r_j)$       $\triangleright$ Max Tanimoto similarity to molecules in $\mathcal{R}$
7:         $\text{MMR}_i \leftarrow \lambda \cdot \sigma(p_i) - (1-\lambda) \cdot \max \text{Sim}_i$       $\triangleright \sigma$: sigmoid function
8:     **end for**
9:     Select $c^* \leftarrow \arg\max_{c_i \in \mathcal{C}} \text{MMR}_i$
10:    $\mathcal{R} \leftarrow \mathcal{R} \cup \{c^*\}, \mathcal{C} \leftarrow \mathcal{C} \setminus \{c^*\}$
11: **end while**

---

active percentages ($< 1\%$). Each WelQrate dataset possesses ten split schemes for robust evaluation. Five random splits adopt a nested cross-validation strategy to allow every molecule in the dataset to be evaluated in the test set. Five scaffold splits with different random seeds separate molecules with different core structures to test models' ability to predict activity across structural variations. More detailed discussion on the random vs scaffold splits is in Appendix A.3 We fine-tuned different methods on each dataset's split and reported the aggregated result for random and scaffold splits, respectively. For this work, we select five representative datasets, shown in Appendix A.1, that cover all therapeutic target types in the WelQrate dataset for evaluation.

**Evaluation Metrics**. Virtual screening focuses on early recognition performance because identifying active compounds within the top-ranked predictions is critical for efficient resource allocation in drug discovery campaigns. To evaluate ranking quality comprehensively, we follow those recommended in WelQrate, more specifically we employ four complementary metrics:

- **logAUC**$_{[0.001, 0.1]}$ measures logarithmic area under the receiver-operating-characteristic curve at false positive rates between [0.001, 0.1] (Liu et al., 2023b; Golkov et al., 2020).

- **BEDROC** ranges from 0 to 1, where a score closer to 1 indicates better performance in recognizing active compounds early in the list (Pearlman & Charifson, 2001).

- **EF**$_{100}$ measures how well a screening method can increase the proportion of active compounds in a selection set with the top 100 compounds, compared to a random selection set (Halgren et al., 2004)

- **DCG**$_{100}$ aims to penalize a true active molecule appearing lower in the selection set by logarithmically reducing the relevance value proportional to the predicted rank of the compound within the top 100 predictions (Järvelin & Kekäläinen, 2017).

We present the formal definitions for each of the above along with more detailed justification for their use in benchmarking in Appendix A.4. Furthermore, we include a novel scaffold diversity metric **SD**$_{100}$ designed to quantify the dissimilarity between molecular scaffolds through pairwise Tanimoto similarity of ECFP fingerprints:

$$\text{SD}_{100} = 1 - \frac{2}{k(k-1)} \sum_{i=1}^{k-1} \sum_{j=i+1}^{k} \text{Sim}(s_i, s_j), \quad k = 100 \tag{2}$$

where $s_i$ and $s_j$ represent molecular scaffolds in top 100 prediction set.

**Baselines and implementation**. We implement various graph neural networks (GNNs) as backbone models to demonstrate the generalizability of **ScaffAug**, including GCN(Kipf & Welling, 2017), GAT(Veličković et al., 2018), and DeepGIN(Xu et al., 2019). We also compare three recent graph augmentation methods: FLAG(Kong et al., 2022), GREA(Liu et al., 2022), and DCT(Liu et al., 2023a), which show potent performance on molecule property prediction. To ensure fair comparison on the severely class-imbalanced WelQrate dataset, we apply an Imbalanced Sampler for **ScaffAug** and all baseline methods. Therefore, we mark the baseline with os, indicating that oversampling of the minor class is used. Each baseline is carefully fine-tuned based on the hyperparameter pool given by its original paper and code with the Optuna package(Akiba et al., 2019). During training, we use

Table 1: Experimental results of all methods across five datasets under random and scaffold splits. The top method for each dataset and data split is highlighted in **bold**. Standard deviation is calculated across five splits for either random or scaffold, with each split including three runs with different random seeds.

| Method / Metric | AID1798 | | AID2689 | | AID463087 | | AID485290 | | AID488997 | |
|---|---|---|---|---|---|---|---|---|---|---|
| | Random | Scaffold | Random | Scaffold | Random | Scaffold | Random | Scaffold | Random | Scaffold |
| **logAUC ↑** | | | | | | | | | | |
| FLAG | 0.167±0.07 | 0.046±0.02 | 0.448±0.06 | 0.302±0.06 | 0.341±0.03 | 0.250±0.07 | 0.236±0.03 | 0.221±0.03 | 0.312±0.09 | 0.182±0.05 |
| GREA | 0.156±0.06 | 0.056±0.02 | 0.426±0.07 | 0.252±0.08 | 0.405±0.02 | 0.312±0.05 | 0.266±0.03 | 0.230±0.03 | 0.328±0.06 | 0.182±0.04 |
| InstructMol | 0.153±0.07 | 0.060±0.01 | 0.381±0.08 | 0.199±0.07 | 0.371±0.04 | 0.245±0.02 | 0.208±0.04 | 0.179±0.04 | 0.313±0.05 | **0.249±0.11** |
| DCT | 0.154±0.07 | 0.070±0.03 | 0.437±0.06 | 0.331±0.09 | 0.430±0.03 | **0.347±0.03** | 0.271±0.03 | 0.234±0.03 | 0.319±0.06 | 0.207±0.06 |
| ScaffAug$_{GCN}$ | 0.112±0.03 | 0.063±0.03 | 0.275±0.07 | 0.174±0.05 | 0.349±0.02 | 0.299±0.05 | 0.190±0.03 | 0.167±0.02 | 0.272±0.08 | 0.137±0.04 |
| ScaffAug$_{GAT}$ | 0.118±0.03 | 0.056±0.02 | 0.325±0.06 | 0.199±0.06 | 0.359±0.02 | 0.291±0.06 | 0.202±0.03 | 0.164±0.02 | 0.250±0.08 | 0.141±0.07 |
| ScaffAug$_{DeepGIN}$ | **0.197±0.07** | **0.078±0.02** | **0.492±0.05** | **0.367±0.08** | **0.433±0.02** | 0.341±0.05 | **0.289±0.02** | **0.270±0.03** | **0.374±0.08** | 0.234±0.02 |
| **BEDROC ↑** | | | | | | | | | | |
| FLAG | 0.250±0.08 | 0.099±0.03 | 0.557±0.06 | 0.439±0.06 | 0.492±0.03 | 0.403±0.07 | 0.348±0.03 | 0.334±0.04 | 0.414±0.10 | 0.295±0.07 |
| GREA | 0.237±0.06 | 0.114±0.03 | 0.552±0.08 | 0.374±0.08 | 0.561±0.04 | 0.480±0.04 | 0.376±0.04 | 0.346±0.04 | 0.413±0.08 | 0.262±0.05 |
| InstructMol | 0.228±0.08 | 0.131±0.02 | 0.495±0.08 | 0.348±0.08 | 0.519±0.04 | 0.412±0.02 | 0.308±0.04 | 0.290±0.04 | 0.401±0.06 | **0.355±0.11** |
| DCT | 0.225±0.07 | 0.132±0.05 | 0.569±0.07 | 0.466±0.10 | **0.585±0.03** | **0.521±0.02** | 0.396±0.03 | 0.353±0.03 | 0.404±0.07 | 0.299±0.08 |
| ScaffAug$_{GCN}$ | 0.201±0.03 | 0.129±0.04 | 0.411±0.06 | 0.289±0.07 | 0.503±0.02 | 0.469±0.05 | 0.303±0.04 | 0.271±0.03 | 0.369±0.09 | 0.218±0.04 |
| ScaffAug$_{GAT}$ | 0.206±0.03 | 0.121±0.04 | 0.454±0.07 | 0.302±0.08 | 0.512±0.02 | 0.455±0.06 | 0.306±0.04 | 0.268±0.03 | 0.332±0.09 | 0.217±0.08 |
| ScaffAug$_{DeepGIN}$ | **0.277±0.07** | **0.140±0.03** | **0.598±0.06** | **0.509±0.10** | 0.580±0.02 | 0.505±0.04 | **0.403±0.02** | **0.392±0.03** | **0.458±0.08** | 0.318±0.04 |
| **EF$_{100}$ ↑** | | | | | | | | | | |
| FLAG | 16.667±10.20 | 2.337±3.00 | 152.237±40.69 | 59.009±45.12 | 39.506±4.56 | 25.781±15.57 | 47.996±14.70 | 42.563±11.06 | 84.897±41.50 | 24.013±20.29 |
| GREA | 15.515±9.58 | 2.647±3.12 | 137.014±50.56 | 58.137±45.01 | 49.774±4.70 | 33.543±10.52 | 61.068±15.12 | 40.847±14.43 | 109.269±30.36 | 47.100±25.73 |
| InstructMol | 16.273±9.44 | 4.443±3.83 | 126.854±43.94 | 28.970±43.07 | 47.790±7.76 | 21.917±4.29 | 48.236±14.52 | 28.785±7.34 | 100.324±20.52 | 53.919±35.16 |
| DCT | 16.274±9.43 | 5.361±3.24 | 128.556±42.31 | **85.518±75.05** | 54.563±6.62 | 36.773±10.74 | 55.349±17.26 | 44.050±14.81 | 109.218±38.89 | 49.083±26.33 |
| ScaffAug$_{GCN}$ | 8.150±5.31 | 3.757±4.29 | 43.980±31.03 | 28.614±34.43 | 41.759±5.03 | 29.343±9.81 | 29.435±13.47 | 27.652±11.29 | 70.179±29.66 | 28.665±18.52 |
| ScaffAug$_{GAT}$ | 7.982±4.95 | 2.210±2.55 | 82.885±35.19 | 33.068±27.57 | 43.418±5.69 | 29.283±11.55 | 37.739±13.31 | 22.901±12.20 | 72.567±33.65 | 21.845±21.90 |
| ScaffAug$_{DeepGIN}$ | **22.147±8.61** | **5.617±2.78** | **177.610±48.90** | 68.867±37.58 | **57.396±4.60** | **39.348±8.64** | **67.777±22.69** | **53.557±14.64** | **135.413±48.33** | **69.754±23.84** |
| **DCG$_{100}$ ↑** | | | | | | | | | | |
| FLAG | 1.153±0.84 | 0.116±0.15 | 1.278±0.43 | 0.616±0.75 | 6.646±0.89 | 4.034±2.95 | 2.492±0.96 | 2.207±0.56 | 1.470±0.71 | 0.413±0.36 |
| GREA | 1.093±0.81 | 0.175±0.25 | 1.605±0.91 | 0.638±0.59 | 8.558±0.83 | 5.496±2.35 | 3.167±1.11 | 1.796±0.65 | 2.447±0.67 | 1.127±0.66 |
| InstructMol | 1.217±0.86 | 0.167±0.12 | 1.418±0.84 | 0.185±0.27 | 8.411±0.86 | 3.850±1.08 | 2.405±0.79 | 1.212±0.30 | 2.107±0.47 | 1.432±0.85 |
| DCT | 1.356±0.96 | 0.273±0.17 | 1.382±0.67 | **0.854±0.88** | 9.317±1.26 | 5.887±2.02 | 2.983±0.86 | 2.323±1.10 | 2.360±0.71 | 1.217±0.73 |
| ScaffAug$_{GCN}$ | 0.540±0.40 | 0.157±0.19 | 0.434±0.33 | 0.205±0.30 | 7.136±0.90 | 5.088±2.28 | 1.731±0.98 | 1.422±0.52 | 1.451±0.79 | 0.579±0.41 |
| ScaffAug$_{GAT}$ | 0.513±0.33 | 0.089±0.11 | 0.695±0.36 | 0.275±0.28 | 7.348±1.17 | 4.870±2.51 | 1.925±1.08 | 1.220±0.73 | 1.459±0.80 | 0.363±0.38 |
| ScaffAug$_{DeepGIN}$ | **1.705±0.97** | **0.397±0.25** | **1.628±0.63** | 0.574±0.50 | **9.393±1.04** | **6.296±2.18** | **3.494±1.33** | **2.989±0.82** | **3.113±0.98** | **1.764±0.88** |

BEDROC as the validation metric due to its stability, which helps produce test results with lower variance. The epoch with the highest BEDROC score on the validation set is selected for the final testing evaluation. We conduct three testing evaluations with different random seeds for each split scheme and aggregate results for the random and scaffold splitting strategies separately.

## 4.2 Experimental Results

**Virtual Screening Performance**. Table 1 presents a thorough evaluation of all methods across five WelQrate datasets using four virtual screening metrics, with results separated for random splits and scaffold splits schemes. ScaffAug$_{DeepGIN}$ consistently achieves the best performance, ranking first across almost all experimental conditions for four metrics. DCT performs better than other graph augmentation baselines, particularly on the AID463087 dataset, likely because it can generate valid molecular graphs. However, DCT(Liu et al., 2023a) only generates new molecules based on those where the virtual screening model already shows high confidence. This approach may not work well with distinct active molecules that are structurally different from clusters of similar active molecules in the training set. In contrast, FLAG(Kong et al., 2022) and GREA(Liu et al., 2022) are general graph augmentation methods that do not consider the validity of molecular graphs, leading to suboptimal performance. These findings motivate us to develop VS-specific augmentation methods to generate valid, drug-like molecules that expand the chemical space of active molecules represented in the training data. By preserving critical scaffolds while introducing structural diversity, ScaffAug enhances the VS model's ability to identify active compounds from novel chemical regions that are not well-represented in the original training set.

**Reranking Analysis**. After evaluating the trained virtual screening model $f_\phi$ on the test set, we derive prediction scores for all test molecules. We only keep the positive scores and sort them, forming the candidate set $\mathcal{C}$ with a maximum size of 500 molecules. The next step applies MMR reranking to $\mathcal{C}$ to obtain the reranked set $\mathcal{R}$. To assess the quality of this reranking, the enrichment factor $EF_{100}$ and the scaffold diversity $SD_{100}$ are calculated on the top 100 subset of the original and reranked set $\mathcal{R}_{100}$. Figure 3 presents the reranking results of ScaffAug$_{DeepGIN}$ across all five WelQrate datasets, showing $SD_{100}$ and $EF_{100}$ for reranked sets with different $\lambda$ values for MMR score calculation. With smaller $\lambda$ values, we gain greater scaffold diversity in the $\mathcal{R}_{100}$ set. Conversely, when $\lambda = 1$, MMR selects compounds based on the original prediction scores without considering structural diversity. Notably, when scaffold diversity is overemphasized ($\lambda \to 0$), $EF_{100}$ typically decreases, indicating

that careful tuning of $\lambda$ is required for each dataset. For the AID1798 and AID2689 scaffold splits, where the model's ability to generalize across structural variations is more severely tested, $EF_{100}$ increases when more scaffold diversity is incorporated into $\mathcal{R}_{100}$. Specifically, for the AID1798 scaffold split, $EF_{100}$ increases by 19.8% while $SD_{100}$ increases by 2%. In other cases, reranking can maintain $EF_{100}$ while increasing some scaffold diversity. This motivates us to practice similar reranking methods focused on structural diversity in the actual virtual screening scenarios.

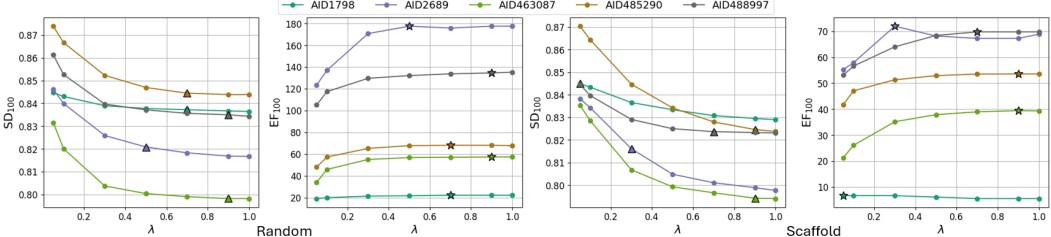

Figure 3: MMR reranking results for ScaffAug$_{DeepGIN}$. $SD_{100}$ and $EF_{100}$ represent scaffold diversity and enrichment factor for the reranked molecular sets, respectively. Note that the original scores before reranking correspond to $\lambda = 1$. The highest $EF_{100}$ score after reranking and its corresponding $SD_{100}$ score at the same $\lambda$ value are highlighted.

**Ablation Study of Augmentation Module**. The augmentation module explores the chemical space around active molecules in the training set, providing additional knowledge for the limited active samples to strengthen model performance. We carefully examine the usefulness of our scaffold-aware sampling (SAS) algorithm on the structural imbalance among active molecules in the training set. As shown in Fig. 5, the active molecules in the training set (AID1798) exhibit a dominant cluster in the central region of the UMAP projection, with over half of the active molecules concentrated in this area, while the remaining molecules have limited presence in peripheral chemical regions. We compare SAS with the uniform actives (UA) sampling algorithm, which selects each active scaffold uniformly with equal probability in the augmentation module. With UA sampling, the generated

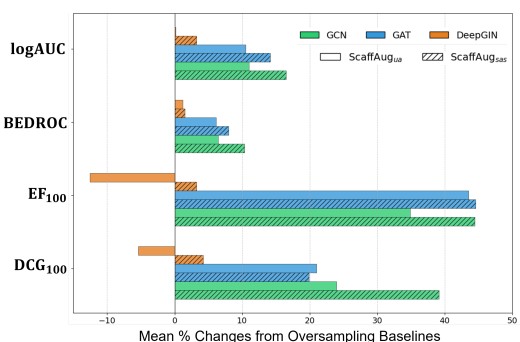

Figure 4: Comparison of different models' oversampling baseline with ScaffAug variants using UA and SAS. Mean percentage change across five datasets and both split types.

molecules maintain a similar distribution pattern to the original training data, with most molecules clustering in the central region and leaving peripheral chemical space under-explored, as evidenced by molecular counts reaching over 250 in the central region while maintaining near-zero representation in peripheral areas. This concentration bias makes the identification of novel active molecules in peripheral areas challenging, as the model receives insufficient training examples from these regions. In contrast, SAS addresses this structural imbalance by assigning higher sampling weights to underrepresented scaffold structures when constructing the scaffold library for molecule generation. The resulting SAS-generated molecules demonstrate improved spatial distribution across the UMAP projection, with more uniform coverage of peripheral chemical regions that were undersampled by UA. This enhanced coverage results in a G-DSA dataset that better represents the full chemical diversity around active compounds in the training set, providing more balanced training examples across different structural motifs and chemical space regions.

Figure 4 compares the percentage change of scaffold-aware sampling (SAS) and uniform actives (UA) variants relative to the oversampling (os) baseline. ScaffAug with SAS consistently outperforms both the oversampling baseline and the UA variant across all evaluation metrics, with this improvement pattern observed across different model architectures. This consistency indicates the effectiveness and robustness of the scaffold-aware generative augmentation framework for virtual screening tasks.

The SAS approach achieves particularly substantial improvements in $EF_{100}$ and $DCG_{100}$ metrics, with performance gains reaching approximately 45% and 38%, respectively, in optimal cases. These enrichment-based metrics are critical indicators of early retrieval performance in drug discovery workflows. The UA variant shows markedly smaller improvements for these same metrics and, more significantly, performs worse than the oversampling baseline for $EF_{100}$ (declining over 10%) and $DCG_{100}$ metrics when applied to certain model architectures. This performance degradation occurs because uniform sampling of active compounds worsens structural imbalance by generating additional molecules from already over-represented scaffold clusters. The resulting bias toward dominant structural patterns impairs the model's ability to generalize to structurally distinct molecules during evaluation. These results underscore the importance of scaffold-aware sampling strategies that actively increase scaffold diversity among active compounds in the training data, rather than simply augmenting the total number of active samples.

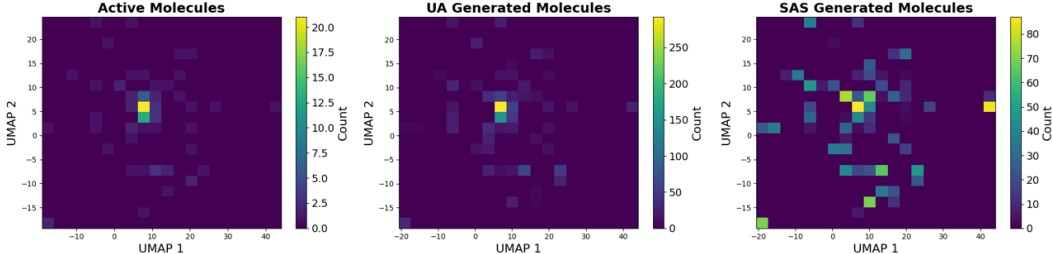

Figure 5: Sampling algorithms affect molecular diversity. We compare the chemical descriptors of original active molecules from the training set, molecules generated using the uniform actives (UA) sampling algorithm, and molecules generated using the scaffold-aware sampling (SAS) algorithm. We convert all molecules to ECFP fingerprints and project them to the same 2D space via UMAP. We then display the number of molecules in each grid, showing the diversity level in different regions of chemical space.

## 5 LIMITATIONS AND FUTURE WORK

Despite ScaffAug's promising performance, some limitations warrant discussion. First, while our generative approach produces chemically valid molecules, it does not guarantee synthetic accessibility or practical feasibility. This gap between computational generation and practical synthesis represents a significant challenge when applying ScaffAug for actual drug discovery campaigns. Second, as a ligand-based virtual screening approach, ScaffAug neglects protein target structures, potentially reducing effectiveness compared to structure-based methods. Future work could address this limitation by integrating protein structure into the generative process, leveraging protein-aware diffusion models(Nobrega et al., 2022; Schneuing et al., 2022) to generate synthetic compounds with a greater likelihood of target activity. Such extensions would enable the framework to benefit from both ligand-based efficiency and structure-based accuracy, potentially yielding even more powerful virtual screening capabilities. Additionally, while our current approach utilizes pseudo-labeling for safely integrating synthetic data, alternative semi-supervised learning methods could be explored to combine unlabeled synthetic data with labeled examples more effectively.

## 6 CONCLUSION

In this work, we introduced **ScaffAug**, a novel scaffold-aware framework that addresses three major challenges in virtual screening: class imbalance, structural imbalance among active molecules, and limited scaffold diversity in top predictions. Our augmentation module uses scaffold-aware sampling and graph diffusion models to generate synthetic molecules that preserve critical scaffolds, the self-training module safely incorporates these molecules through confidence-based pseudo-labeling, and the reranking module increases scaffold diversity while maintaining hit identification performance. Extensive experiments across five therapeutic target classes demonstrate that ScaffAug consistently outperforms existing baselines on multiple evaluation metrics for both random and scaffold splits. ScaffAug represents a significant advancement in computational drug discovery by

leveraging generative models to enhance virtual screening, particularly in early-stage campaigns where identifying structurally diverse active compounds is critical.

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

# A APPENDIX

## A.1 DATASET DETAILS

The WelQrateLiu et al. (2024b) collection establishes a high-quality benchmark through the rigorous curation of experimental data sourced from PubChem. The suite comprises nine distinct datasets that encompass a variety of critical drug targets, all characterized by the challenging, low active-to-inactive ratios typical of realistic drug discovery scenarios.

Table 2: Details of the five datasets used for benchmarking.

| Target Class | BioAssay ID (AID) | Target | Compound Type | Number of Compounds | Number of Actives | Percent Active |
|---|---|---|---|---|---|---|
| G Protein-Coupled Receptor | 1798 | M1 Muscarinic Receptor | Allosteric Agonist | 60,706 | 164 | 0.270% |
| Ion Channel | 463087 | Cav3 T-type Calcium Channel | Inhibitor | 95,650 | 652 | 0.682% |
| Transporter | 488997 | Choline Transporter | Inhibitor | 288,564 | 236 | 0.082% |
| Kinase | 2689 | Serine Threonine Kinase 33 | Inhibitor | 304,475 | 120 | 0.039% |
| Enzyme | 485290 | Tyrosyl-DNA Phosphodiesterase | Inhibitor | 281,146 | 586 | 0.208% |

## A.2 BASELINES

We implemented the baseline according to their published code and tuned the key hyperparameters

**FLAG** FLAG Kong et al. (2022) (Free Large-scale Adversarial Augmentation on Graphs) iteratively injects small, gradient-based perturbations into node features (while keeping graph structure unchanged) to force the GNN to become robust to feature-level noise. It is lightweight (incurs minimal overhead), compatible with arbitrary GNN backbones, and scales well to large graph datasets.

**GREA** GREA (Liu et al., 2022) (Graph Rationalization with Environment-based Augmentation) decomposes each graph into a rationale subgraph (the core predictive part) and an environment subgraph, then augments by selectively removing, swapping, or perturbing the environment while preserving the rationale. This encourages the model to focus on essential structural components under variant contexts, without arbitrarily altering the core substructure.

**DCT** DCT Liu et al. (2023a) uses a diffusion model pretrained on large unlabeled graph collections to learn the graph distribution, then performs task-conditioned augmentation: for each labeled example, it perturbs the graph via forward diffusion and then reverses it under dual objectives of preserving the original label and promoting structural diversity. Augmented graphs thus carry minimal sufficient knowledge from the unlabeled domain and enrich the training set.

**InstructMol** InstructMol (Cao et al., 2025) introduces a flexible semi-supervised learning framework for molecular property prediction that splits the workflow into two models: a target model that assigns pseudo-labels to large unlabeled molecular datasets, and an instructor model that estimates the reliability (confidence scores) of those pseudo-labels. The target model is then trained on both labeled and pseudo-labeled data using a loss re-weighting strategy guided by the instructor's confidence scores, enabling effective utilization of unlabeled data without relying on domain transfer between disjoint pre-training and fine-tuning stages.

## A.3 DATA SPLITS

Conventional cross-validation evaluates only part of the dataset while tuning hyperparameters on the remainder, leaving portions untested. Nested cross-validation achieves complete coverage, but requires extensive computation due to repeated training across multiple splits. To establish a practical protocol for large-scale benchmarks, we adopt a simplified variant: every sample is eventually tested, but the inner loop is restricted to a single split, where the validation fold directly precedes the test

fold. This design ensures full test coverage while limiting the total number of trained models to five, offering a scalable compromise between robustness and efficiency.

For scaffold-based splits, we propose a benchmark standard that explicitly supports scaffold hopping, a central objective in drug discovery for generating structurally novel compounds with improved properties and patentability Hu et al. (2017). We use Bemis–Murcko (BM) scaffolds Bemis & Murcko (1996) with a 3:1:1 training:validation:test ratio, assigning any scaffold bin exceeding 10% of the dataset to the training set. This ensures scaffold diversity is preserved across all splits.

## A.4 EVALUATION METRICS DETAILS

**Logarithmic Receiver Operating Characteristic Area Under the Curve within [0.001, 0.1] (logAUC$_{[0.001,0.1]}$)**

The ranged logAUC metric (Mysinger & Shoichet (2010)) captures performance in the low false-positive regime, emphasizing the left-hand portion of the ROC curve. Since the optimal threshold is unknown, the metric integrates the area under the ROC curve across a restricted FPR interval. Taking the logarithm accentuates contributions from extremely small FPR values. Following prior studies (Liu et al. (2023b); Golkov et al. (2020)), we adopt the interval [0.001, 0.1]. A perfect predictor achieves logAUC$_{[0.001,0.1]} = 1$, while random guessing gives a value of approximately 0.0215, computed as:

$$\frac{\int_{0.001}^{0.1} x \, \mathrm{d} \log_{10} x}{\int_{0.001}^{0.1} 1 \, \mathrm{d} \log_{10} x} = \frac{\int_{-3}^{-1} 10^u \mathrm{d}u}{\int_{-3}^{-1} 1 \mathrm{d}u} \approx 0.0215$$

**Boltzmann-Enhanced Discrimination of Receiver Operating Characteristic (BEDROC)**

BEDROC (Pearlman & Charifson (2001)) ranges between 0 and 1, quantifying a model's ability to prioritize active molecules early in a ranked list. It is derived from the Robust Initial Enhancement ($RIE$), normalized using its theoretical minimum $RIE_{min}$ (all actives at the end of the list) and maximum $RIE_{max}$ (all actives at the top). The definitions are:

$$RIE = \frac{\frac{1}{n} \sum_{i=1}^{n} e^{-\alpha x_i}}{\frac{1}{N} \left( \frac{1-e^{-\alpha}}{e^{\alpha/N}-1} \right)}$$

$$RIE_{max} = \frac{1 - e^{-\alpha R_a}}{R_a \left(1 - e^{-\alpha}\right)}, \quad RIE_{min} = \frac{1 - e^{\alpha R_a}}{R_a \left(1 - e^{\alpha}\right)}$$

Here $n$ and $N$ denote the number of actives and total compounds, respectively. $x_i = r_i/N$ is the normalized rank of the $i$th active, $R_a = n/N$ is the active ratio, and $\alpha$ controls sensitivity to early enrichment. As recommended in the original paper, we use $\alpha = 20$. The final BEDROC score is:

$$BEDROC = \frac{RIE - RIE_{min}}{RIE_{max} - RIE_{min}} = \frac{\sum_{i=1}^{n} -e^{r_i/N}}{\frac{n}{N} \left( \frac{1-e^{-\alpha}}{e^{\alpha/N}-1} \right)} \times \frac{R_a \sinh(\alpha/2)}{\cosh(\alpha/2) - \cosh(\alpha/2 - \alpha R_a)} + \frac{1}{1 - e^{\alpha(1-R_a)}}.$$

**Enrichment Factor at Top 100 (EF$_{100}$)**

The enrichment factor (Halgren et al. (2004)) evaluates how effectively a method increases the concentration of true actives within a top-ranked subset relative to random selection. Using the top 100 predictions as the cutoff, it is defined as:

$$EF_{100} = \frac{n_{100}/N_{100}}{n/N}$$

where $n_{100}$ is the number of actives among the top 100 predicted molecules, $N_{100} = 100$, $n$ is the total number of actives, and $N$ is the total dataset size. By construction, random selection yields EF$_{100} = 1$, while the score falls to 0 if no actives appear in the top 100.

**Discounted Cumulative Gain at 100 (DCG$_{100}$)**

Discounted Cumulative Gain (Järvelin & Kekäläinen (2017)) measures ranking quality by penalizing relevant items that appear deeper in the list. First, the simpler Cumulative Gain (CG) counts the actives in the top 100:

$$CG_{100} = \sum_{i=1}^{100} y_i$$

with $y_i = 1$ if compound $i$ is active, and 0 otherwise. Unlike CG, DCG discounts relevance by rank position, rewarding methods that place actives earlier:

$$DCG_{100} = \sum_{i=1}^{100} \frac{y_i}{\log_2(i+1)}.$$

## A.5 Scaffold Extension via DiGress

This section details the conditional sampling procedure we use to extend a given scaffold with the DiGress Vignac et al. (2022) backbone. Starting from a scaffold graph $G_{\text{scaffold}}$ and its mask $m$, we sample a target size $n > n'$, draw $G_T \sim q_X(n) \times q_E(n)$, and run $T$ reverse steps while fixing the masked scaffold and updating the remaining nodes and edges with the model posteriors; we set $T = 50$. The algorithm returns a candidate molecule, and chemically valid outputs are retained to build the G-DSA set for downstream self-training.

---

**Algorithm 3** Scaffold Extension via DiGress

---

1: **Input**: the scaffold $G_{\text{scaffold}}$ with $n'$ atoms and mask $m$
2: Sample $n > n'$ from the training data distribution
3: Sample $G^T \sim q_X(n) \times q_E(n)$          ▷ Random graph
4: **for** $t = T$ to 1 **do**
5:      $G^t = m \odot G_{\text{scaffold}} + (1-m) \odot G^t$
6:      $\hat{p}^X, \hat{p}^E \leftarrow \phi_\theta(G^t)$          ▷ Forward pass
7:      $p_\theta(x_i^{t-1}|G^t) \leftarrow \sum_x q(x_i^{t-1}|x_i = x, x_i^t)\hat{p}_i^X(x)$    $i \in 1, \ldots, n$          ▷ Posterior
8:      $p_\theta(e_{ij}^{t-1}|G^t) \leftarrow \sum_e q(e_{ij}^{t-1}|e_{ij} = e, e_{ij}^t)\hat{p}_{ij}^E(e)$    $i, j \in 1, \ldots, n$          ▷ Posterior
9:      $G^{t-1} \sim \prod_i p_\theta(x_i^{t-1}|G^t) \prod_{ij} p_\theta(e_{ij}^{t-1}|G^t)$          ▷ Categorical distr.
10: **end for**
11: **return** $G^0$

---

## A.6 SELF-TRAINING ALGORITHM

This section presents the confidence-based self-training loop that integrates unlabeled G-DSA molecules with labeled data. After a warm-up of $E_{\text{start}}$ epochs on $D$, the model generates pseudo-labels for $D'$ every $E_{\text{freq}}$ epochs; molecules with confidence above threshold $\tau$ form $D'_{\text{conf}}$. We then train on $D \cup D'_{\text{conf}}$ with class balancing via oversampling, optimize with BCEWithLogitsLoss under a polynomial learning-rate schedule, and select models using BEDROC on the validation set.

---

**Algorithm 4** Self-Training with Confidence-Based Pseudo-Labeling

---

1: **procedure** SELFTRAINING(f_theta, D, D', D_valid, D_test, conf_threshold, start_epoch, freq)
2:     best_BEDROC $\leftarrow -\infty$, best_epoch $\leftarrow 0$, f_theta_best $\leftarrow$ f_theta
3:     Initialize PolynomialLR scheduler
4:     **for** epoch = 0 to num_epochs **do**
5:         **if** epoch < start_epoch **then**
6:             D_train $\leftarrow$ D                                    ▷ Warm-up phase with original labeled data
7:         **else if** epoch $\geq$ start_epoch **and** epoch mod freq = 0 **then**
8:             D'_confident $\leftarrow$ GENERATEPSEUDOLABELS(f_theta, D', conf_threshold)
9:             D_train $\leftarrow$ D $\cup$ D'_confident        ▷ Combine original and confident augmented data
10:         **end if**
11:         Apply oversampling to balance active and inactive samples in D_train
12:         f_theta $\leftarrow$ OPTIMIZE(f_theta, D_train, $\nabla$L)                          ▷ L: BCEWithLogitsLoss
13:         Update learning rate with PolynomialLR scheduler
14:         BEDROC_valid $\leftarrow$ EVALUATE(f_theta, D_valid)
15:         **if** BEDROC_valid > best_BEDROC **then**
16:             best_BEDROC $\leftarrow$ BEDROC_valid
17:             best_epoch $\leftarrow$ epoch
18:             f_theta_best $\leftarrow$ f_theta
19:         **end if**
20:     **end for**
21:     test_metrics $\leftarrow$ EVALUATE(f_theta_best, D_test)
22:     **return** test_metrics, f_theta_best
23: **end procedure**

---

## A.7 ABLATION STUDY FOR SELF-TRAINING MODULE

In this section, we demonstrate the effectiveness of the self-training module. We evaluate a variant, ScaffAug$_{da}$, which assigns the active label to all generated molecules and merges them directly with the original dataset. Using a GCN backbone, we observe that this direct augmentation strategy performs worse than naive GCN training. Conversely, the proposed self-training module consistently outperforms the GCN baseline across all five datasets under both random and scaffold splits. Note that we apply oversampling of active molecules during training to all experiment settings because of the inherent low active rate of the WelQrate datasets.

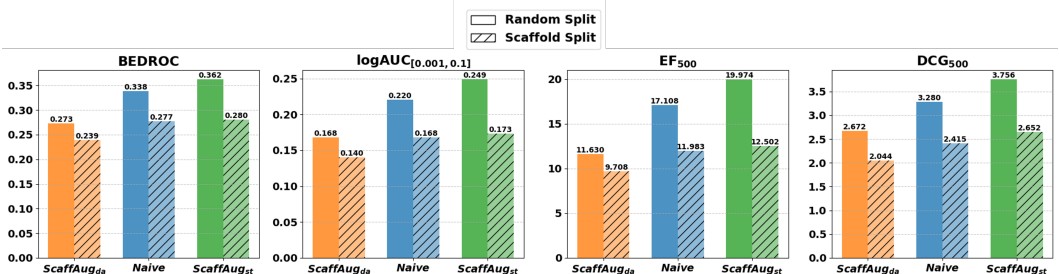

Figure 6: Comparison between self-training and direct augmentation

Table 3: Experimental Result of GCN and its ScaffAug's self-training and direct augmentation variants

| Method / Metric | AID1798 | | AID2689 | | AID463087 | | AID485290 | | AID488997 | |
|---|---|---|---|---|---|---|---|---|---|---|
| | Random | Scaffold | Random | Scaffold | Random | Scaffold | Random | Scaffold | Random | Scaffold |
| **logAUC ↑** | | | | | | | | | | |
| GCN | 0.108±0.022 | 0.056±0.017 | 0.237±0.028 | 0.160±0.024 | 0.303±0.013 | 0.265±0.019 | 0.167±0.006 | 0.164±0.010 | 0.164±0.031 | 0.115±0.035 |
| ScaffAug$_{da}$ | 0.095±0.009 | 0.060±0.009 | 0.138±0.012 | 0.133±0.015 | 0.285±0.008 | 0.264±0.007 | 0.098±0.008 | 0.094±0.004 | 0.151±0.016 | 0.141±0.010 |
| ScaffAug$_{st}$ | 0.109±0.018 | 0.043±0.017 | 0.272±0.031 | 0.169±0.017 | 0.356±0.014 | 0.293±0.019 | 0.174±0.011 | 0.166±0.020 | 0.237±0.033 | 0.108±0.014 |
| **BEDROC↑** | | | | | | | | | | |
| GCN | 0.202±0.029 | 0.118±0.021 | 0.370±0.033 | 0.270±0.030 | 0.474±0.013 | 0.434±0.021 | 0.274±0.011 | 0.270±0.012 | 0.247±0.038 | 0.199±0.046 |
| ScaffAug$_{da}$ | 0.184±0.013 | 0.132±0.012 | 0.254±0.019 | 0.248±0.022 | 0.449±0.009 | 0.429±0.008 | 0.181±0.010 | 0.180±0.007 | 0.225±0.027 | 0.225±0.009 |
| ScaffAug$_{st}$ | 0.198±0.020 | 0.097±0.029 | 0.403±0.036 | 0.288±0.024 | 0.509±0.014 | 0.454±0.021 | 0.284±0.012 | 0.273±0.019 | 0.326±0.035 | 0.188±0.018 |
| **EF$_{100}$↑** | | | | | | | | | | |
| GCN | 7.672±3.918 | 3.229±2.322 | 38.905±16.047 | 15.708±21.619 | 30.317±3.143 | 24.651±4.570 | 23.682±5.255 | 24.131±6.709 | 34.297±21.258 | 13.404±13.871 |
| ScaffAug$_{da}$ | 6.070±3.665 | 1.817±1.388 | 25.373±9.265 | 12.996±10.394 | 28.746±1.899 | 25.288±1.929 | 10.557±5.109 | 9.719±4.226 | 38.407±11.413 | 19.853±14.100 |
| ScaffAug$_{st}$ | 7.756±3.918 | 2.027±1.363 | 52.437±27.012 | 18.690±13.254 | 41.935±4.543 | 30.902±3.786 | 23.348±13.917 | 23.259±15.113 | 66.939±23.045 | 11.623±4.853 |
| **DCG$_{100}$↑** | | | | | | | | | | |
| GCN | 0.463±0.207 | 0.172±0.148 | 0.304±0.107 | 0.130±0.178 | 5.332±0.596 | 3.666±0.605 | 1.298±0.265 | 1.202±0.248 | 0.671±0.522 | 0.211±0.212 |
| ScaffAug$_{da}$ | 0.377±0.183 | 0.154±0.111 | 0.228±0.094 | 0.083±0.079 | 5.597±0.443 | 4.431±0.235 | 0.597±0.254 | 0.526±0.313 | 0.701±0.260 | 0.298±0.200 |
| ScaffAug$_{st}$ | 0.479±0.202 | 0.081±0.053 | 0.490±0.331 | 0.125±0.082 | 7.327±0.625 | 5.203±0.642 | 1.209±0.725 | 1.084±0.488 | 1.286±0.660 | 0.213±0.093 |

## A.8 VISUALIZATION OF AUGMENTATION MODULE

### A.8.1 GENERATED SCAFFOLDS

In the figure below, we show the sampled active scaffold used for augmentation. The remaining scaffolds are novel and do not exist in the original active space of the training set. They are ranked by their similarity scores relative to the sampled scaffold. Although the scaffold is fixed during scaffold-conditioned generation, we can still increase scaffold diversity across the generated active molecules by editing this core structure.

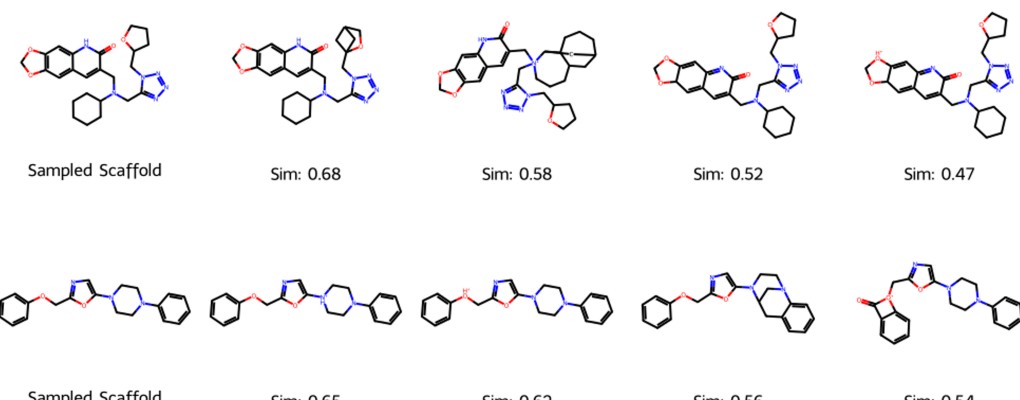

Figure 7: The sampled scaffolds used for augmentation and the novel scaffolds generated in the augmentation module

### A.8.2 SCAFFOLD EXTENSION STEPS

Below, we visualize the scaffold-conditioned generation process, corresponding to the reverse step described in Alg. 3. We use a maximum of 50 reverse steps. Initially, the unknown region consists of noise sampled from the unlabeled data distribution. As the reverse steps progress, this noise gradually transforms into meaningful substructures around the scaffold, eventually forming valid molecules for augmentation.

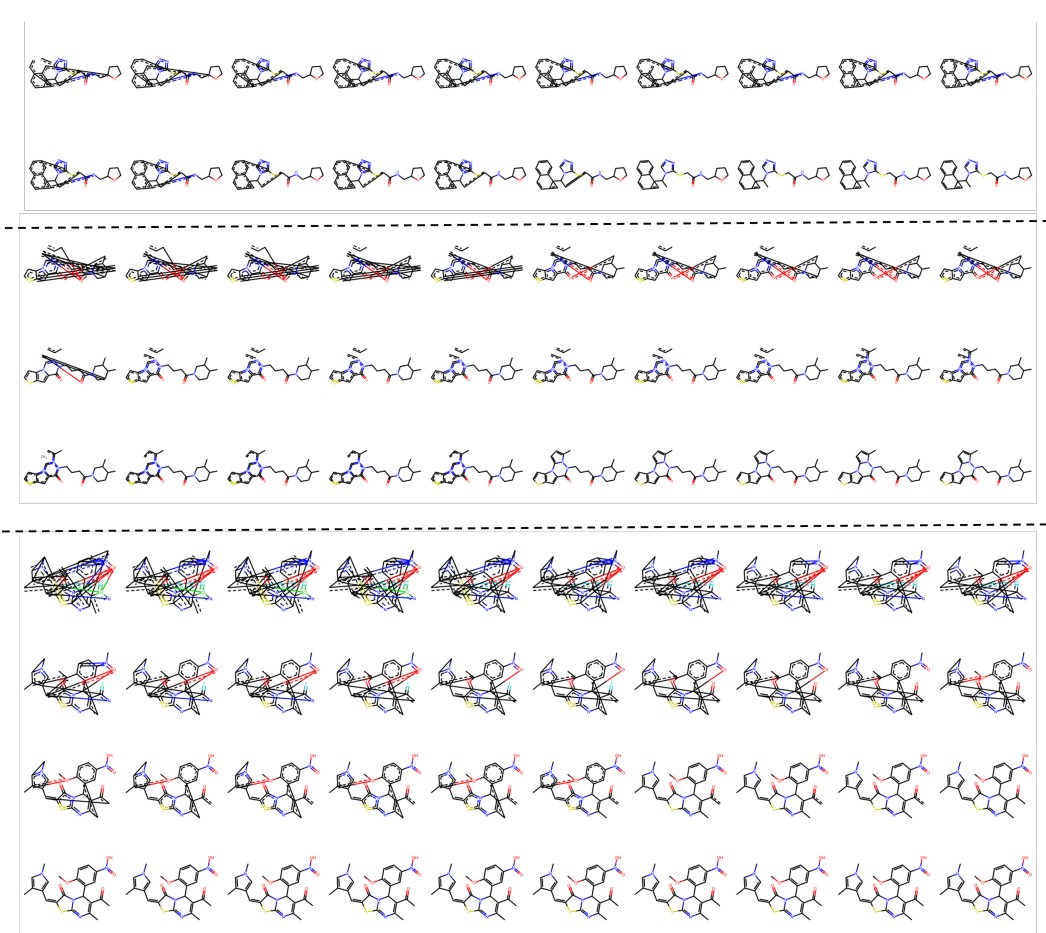

Figure 8: Reverse Steps of Scaffold-conditioned Generation

