# OpenReview forum: "Scaffold-Aware Generative Augmentation and Reranking for Enhanced Virtual Screening"
_ICLR.cc/2026/Conference — ICLR 2026 Conference Desk Rejected Submission_

### Official Review · Reviewer_zpNN · 2025-10-29

**Soundness:** 3
**Presentation:** 3
**Contribution:** 2
**Rating:** 2
**Confidence:** 5

**Summary:**

This paper addresses virtual screening challenges through scaffold-aware augmentation (oversampling underrepresented molecular scaffolds and extending them via graph diffusion), self-training with pseudo-labeling, and MMR-based reranking for diversity.

**Strengths:**

1.The key conceptual contribution is distinguishing structural imbalance (underrepresented scaffold families) from class imbalance, with scaffold-aware sampling as a novel solution to oversample rare scaffolds during generation. While individual components are standard techniques, their chemistry-aware integration and the specific focus on balancing scaffold diversity represents a creative domain-specific applications.

2. The paper is overall well-structured with clear problem formulation, visual overview, an visualization showing improved coverage of underrepresented chemical space. Mathematical definitions, algorithms, and the distinction between evaluation splits are all clearly presented.

**Weaknesses:**

1. Overall, the paper seems to be more of an application paper with good empirical results than a methods innovation paper:  combining existing techniques in a chemistry-aware way for virtual screening. It's fundamentally semi-supervised learning + importance sampling with conditional generation and diversity reranking(MMR). Is this enough for ICLR is debatable. While not completely overlapping similar concepts have been used before in different field of molecule prediction tasks (refer to Instructor-inspired Machine Learning for Robust
Molecular Property Prediction (Neurips 2024))

2. No ablation on the three modules, only for Scaffold-aware sampling (SAS) vs Uniform actives (UA) sampling.

3. A clearer articulation why scaffold diversity is critical for virtual screening success can make the paper more convincing

**Questions:**

1. What if you only use augmentation without self-training? Only self-training without reranking? They claim three modules but never test them individually. Does preserving the scaffold actually matter? What if you just generate molecules without fixing scaffolds?

2. How many generated molecules are actually valid/diverse? What's the generation success rate?

3. Compared to Instructor-inspired Machine Learning for Robust Molecular Property Prediction (Neurips 2024), what makes this paper's method novel other than scaffold-aware sampling and mmr-based reranking which, again, is not novel?

---

> ### Author Response · Authors · 2025-11-22
> **Response to Weakness 1**
>
> # Weakness
> > 1. Overall, the paper seems to be more of an application paper with good empirical results than a methods innovation paper: combining existing techniques in a chemistry-aware way for virtual screening. It's fundamentally semi-supervised learning + importance sampling with conditional generation and diversity reranking(MMR). Is this enough for ICLR is debatable. While not completely overlapping, similar concepts have been used before in different field of molecule prediction tasks (refer to Instructor-inspired Machine Learning for Robust Molecular Property Prediction (Neurips 2024))
>
> We respectively disagree that ScaffAug "seems to be an application paper with empirical results" and argue that it provides method-level contributions tailored to real-world ligand-based virtual screening.
>
> - The complete research pipeline from data-centric challenges to method design should be valued. ScaffAug starts from identifying major challenges in realistic HTS datasets: (i) extreme class imbalance, (ii) structural imbalance among active scaffolds, and (iii) the need for discovering diverse active compounds. Scaffold-aware sampling and scaffold extension specifically target (i) and (ii) by over-sampling underrepresented active scaffolds, growing them with a scaffold-conditioned diffusion model, and thus generating new molecules that can enhance prediction models' capability of identifying active molecules in the top list. The simple self-training strategy is used to demonstrate the quality of the augmentation dataset. The MMR reranking module further focuses on (iii) by explicitly trading off prediction scores and scaffold diversity in the final top-k set.
>
> - The scaffold-aware augmentation framework is modular and model-agnostic on both the predictive and generative components. ScaffAug supports multiple GNN backbones and can seamlessly incorporate other generative models, such as SAFE, without modifying the rest of the pipeline. This flexibility makes it a general and easily reusable scaffold-aware augmentation scheme for future developers.
>
> We admit InstructMol is a novel semi-supervised method that leverages unlabeled data to assist molecular property prediction. It requires a target model (f) to predict pseudo-labels and an instructor model $g$ to judge reliability: (1) pretrain $f, g$; (2) $f$ assigns pseudo-labels to all unlabeled data; (3) $g$ predicts confidence scores; (4) $g$ is trained to distinguish real labels from pseudo-labels, and $f$ is trained with a re-weighted loss according to these scores. However, it has several limitations.
>
> * InstructMol does not generalize well to real-world virtual screening datasets, where the class imbalance (active rate < 1%) and structural imbalance among active molecules are severe. We benchmarked InstructMol using the same DeepGIN prediction backbone and ZINC15_250K as unlabeled data with the default hyperparameters. According to the table below, ScaffAug consistently outperforms InstructMol except on AID488997 scaffold splits.
>
> * InstructMol assigns pseudo-labels to a large amount of unlabeled data (250k or 1M) and trains on them with a small labeled dataset (<10k). This wastes computing resources because most unlabeled data are irrelevant or unhelpful for the prediction task. Therefore, InstructMol does not scale well to large labeled datasets like WelQrate, where the size is already up to 300k. This highlights the novelty of ScaffAug: it uses scaffold-aware sampling and scaffold extension to generate small but useful unlabeled data that tackle class and structural imbalance, resulting in increasing the model’s capability to identify novel active molecules in the vast chemical space.

---

> > ### Author Response · Authors · 2025-11-22
> > **Table for Response to Weakness 1**
> >
> > We have included the empirical results of InstructMol in the revised manuscript.
> >
> > | Method / Metric | AID1798 Random | AID1798 Scaffold | AID2689 Random | AID2689 Scaffold | AID463087 Random | AID463087 Scaffold | AID485290 Random | AID485290 Scaffold | AID488997 Random | AID488997 Scaffold |
> > |---|---|---|---|---|---|---|---|---|---|---|
> > | **logAUC** | | | | | | | | | | |
> > | InstructMol | 0.153 $\pm$ 0.07 | 0.060 $\pm$ 0.01 | 0.381 $\pm$ 0.08 | 0.199 $\pm$ 0.07 | 0.371 $\pm$ 0.04 | 0.245 $\pm$ 0.02 | 0.208 $\pm$ 0.04 | 0.179 $\pm$ 0.04 | 0.313 $\pm$ 0.05 | **0.249 $\pm$ 0.11** |
> > | ScaffAug$_{DeepGIN}$ | **0.197 $\pm$ 0.07** | **0.078 $\pm$ 0.02** | **0.492 $\pm$ 0.05** | **0.367 $\pm$ 0.08** | **0.433 $\pm$ 0.02** | **0.341 $\pm$ 0.05** | **0.289 $\pm$ 0.02** | **0.270 $\pm$ 0.03** | **0.374 $\pm$ 0.08** | 0.234 $\pm$ 0.02 |
> > | **BEDROC** | | | | | | | | | | |
> > | InstructMol | 0.228 $\pm$ 0.08 | 0.131 $\pm$ 0.02 | 0.495 $\pm$ 0.08 | 0.348 $\pm$ 0.08 | 0.519 $\pm$ 0.04 | 0.412 $\pm$ 0.02 | 0.308 $\pm$ 0.04 | 0.290 $\pm$ 0.04 | 0.401 $\pm$ 0.06 | **0.355 $\pm$ 0.11** |
> > | ScaffAug$_{DeepGIN}$ | **0.277 $\pm$ 0.07** | **0.140 $\pm$ 0.03** | **0.598 $\pm$ 0.06** | **0.509 $\pm$ 0.10** | **0.580 $\pm$ 0.02** | **0.505 $\pm$ 0.04** | **0.403 $\pm$ 0.02** | **0.392 $\pm$ 0.03** | **0.458 $\pm$ 0.08** | 0.318 $\pm$ 0.04 |
> > | **EF100** | | | | | | | | | | |
> > | InstructMol | 16.273 $\pm$ 9.44 | 4.443 $\pm$ 3.83 | 126.854 $\pm$ 43.94 | 28.970 $\pm$ 43.07 | 47.790 $\pm$ 7.76 | 21.917 $\pm$ 4.29 | 48.236 $\pm$ 14.52 | 28.785 $\pm$ 7.34 | 100.324 $\pm$ 20.52 | 53.919 $\pm$ 35.16 |
> > | ScaffAug$_{DeepGIN}$ | **22.147 $\pm$ 8.61** | **5.617 $\pm$ 2.78** | **177.610 $\pm$ 48.90** | **68.867 $\pm$ 37.58** | **57.396 $\pm$ 4.60** | **39.348 $\pm$ 8.64** | **67.777 $\pm$ 22.69** | **53.557 $\pm$ 14.64** | **135.413 $\pm$ 48.33** | **69.754 $\pm$ 23.84** |
> > | **DCG100** | | | | | | | | | | |
> > | InstructMol | 1.217 $\pm$ 0.86 | 0.167 $\pm$ 0.12 | 1.418 $\pm$ 0.84 | 0.185 $\pm$ 0.27 | 8.411 $\pm$ 0.86 | 3.850 $\pm$ 1.08 | 2.405 $\pm$ 0.79 | 1.212 $\pm$ 0.30 | 2.107 $\pm$ 0.47 | 1.432 $\pm$ 0.85 |
> > | ScaffAug$_{DeepGIN}$ | **1.705 $\pm$ 0.97** | **0.397 $\pm$ 0.25** | **1.628 $\pm$ 0.63** | **0.574 $\pm$ 0.50** | **9.393 $\pm$ 1.04** | **6.296 $\pm$ 2.18** | **3.494 $\pm$ 1.33** | **2.989 $\pm$ 0.82** | **3.113 $\pm$ 0.98** | **1.764 $\pm$ 0.88** |

---

> ### Author Response · Authors · 2025-11-22
> **Response to Weakness 2, 3**
>
> > 2. No ablation on the three modules, only for Scaffold-aware sampling (SAS) vs Uniform actives (UA) sampling.
>
> Thank you for the comment. We actually include the ablation studies for the three modules: augmentation module, self-training module, and reranking module, respectively, in the manuscript.
>
> **The augmentation module**. We first validate the effectiveness of scaffold-aware sampling with **Fig. 4 and Fig. 5**.
> - Figure 4 reports the percentage improvements of our two augmentation variants (SAS and UA) over the baseline across four ranking metrics. The baseline trains on the original dataset while oversampling active molecules during training. For each backbone model (GCN, GAT, and DeepGIN), the baseline is represented by the zero reference line, and the colored bars indicate the relative performance gains achieved by our augmentation strategies. As shown in the figure, $\text{ScaffAug}_\text{sas}$ consistently achieves the largest improvements over the oversampling baseline across nearly all metrics and models.
> - Figure 5 shows that scaffold-aware sampling (SAS) tackles the structural imbalance challenge by expanding coverage of chemical space around actives selectively. Compared with both the original actives and the uniform actives (UA) sampling, SAS-generated molecules fill peripheral and previously underrepresented regions in the chemical space, rather than just duplicating dense regions.
>
> **The self-training module**. We demonstrate the effectiveness of the self-training module by evaluating a variant, $\textrm{ScaffAug}_{da}$, which assigns the active label to all generated molecules and merges them directly with the original dataset. Then, we train the GCN backbone on the augmented dataset. We observe that this direct augmentation strategy performs worse than naïve GCN training. In contrast, the proposed self-training module consistently outperforms the GCN baseline across all five datasets under both random and scaffold splits, as shown in **Figure 6** and **Table 3** in *Appendix A.7*.
>
> **The reranking module.** Note that we do not apply the reranking module for the results reported in Table 1. In **Figure 3**, we study the effect of the reranking module by varying the trade-off parameter $\lambda$ for $\text{ScaffAug}_{\text{DeepGIN}}$. $\lambda = 1.0$ corresponds to ScaffAug without reranking, where molecules are ordered only by the predicted scores, while $\lambda < 1.0$ activates the MMR module and gradually puts more weight on scaffold diversity. As $\lambda$ decreases from $1.0$, SD100 increases consistently, and EF100 stays similar or even improves in several assays, which shows that reranking adds clear value on top of the base model by improving scaffold diversity in the top hits without sacrificing early enrichment.
>
> ---
>
> > 3. A clearer articulation why scaffold diversity is critical for virtual screening success can make the paper more convincing
>
> Thank you for the suggestion. Scaffold diversity is critical for the success of virtual screening for several reasons.
> - Diverse scaffolds help avoid patented or heavily explored chemical spaces, increasing the likelihood of identifying novel, patentable compounds.
> - Alternative scaffolds may offer more favourable properties, such as solubility, membrane permeability, stability, and metabolic profile, which are essential for progressing hits into viable drug candidates.
> - Structurally diverse scaffolds provide more chances to identify compounds with simpler and more feasible synthetic routes, reducing development time and cost.
>
> [1] Brown, Nathan. (2013). Scaffold Hopping in Medicinal Chemistry. 10.1002/9783527665143.ch01.

---

> ### Author Response · Authors · 2025-11-22
> **Response to Questions**
>
> # Question
> > 1. What if you only use augmentation without self-training? Only self-training without reranking? They claim three modules but never test them individually. Does preserving the scaffold actually matter? What if you just generate molecules without fixing scaffolds?
>
> We appreciate the question. We have included the comprehensive ablation analysis of the **augmentation module**, **self-training** module, and **reranking module** in the response to weakness 2.
>
> >  "Does preserving the scaffold actually matter? What if you just generate molecules without fixing scaffolds?"
>
> Yes, preserving the scaffolds from the active molecules is critical in our work. If a scaffold appears in active compounds, we assume it contributes to the binding signal against the target and carries desirable molecular properties such as solubility, membrane permeability, stability, and metabolic profile. Therefore, ScaffAug values and augments the underrepresented scaffolds from the active molecules to reduce the model’s bias to overfit to a small number of dominant scaffolds and increase the overall scaffold diversity of active molecules in the training set.
>
> If we do not fix the scaffolds during generation, it means the generative model performs unconditional generation to sample from the unlabeled data distribution. In this case, the generated molecules drift away from the distribution of active molecules in the dataset, and semi-supervised methods become less useful. InstructMol provides a good example. Although it leverages a large labeled dataset (ZINC15_250K) that spans a broad chemical space and incorporates a carefully designed semi-supervised framework, it still does not outperform ScaffAug, which only relies on a much smaller amount of augmentation data that explicitly preserves active scaffolds.
>
> ---
> > 2. How many generated molecules are actually valid/diverse? What's the generation success rate?
>
> Thank you for raising this question. The validity rate of our trained DiGress model is 0.85, with a uniqueness of 1.0 when sampling 10,000 molecules. Scaffold extension is inherently more challenging, which leads to a lower validity rate. Below, we report the mean validity rate across splits for all datasets. Because reduced validity may affect the effectiveness of scaffold-aware sampling, we iteratively generate molecules for each sampled scaffold until a valid molecule is obtained. **Only valid molecules are included in our augmentation dataset.**
>
> | Dataset   | Validity Rate|
> | --------- | --------------------- |
> | AID1798   | 0.6244                |
> | AID2689   | 0.5314                |
> | AID463087 | 0.6117                |
> | AID485290 | 0.6953                |
> | AID488997 | 0.6251                |
>
> ---
>
> > 3. Compared to Instructor-inspired Machine Learning for Robust Molecular Property Prediction (Neurips 2024), what makes this paper's method **novel** other than scaffold-aware sampling and mmr-based reranking which, again, is not novel?
>
> Thank you for the question. We have included the novelty of ScaffAug in the response to weakness 1.

---

### Official Review · Reviewer_BG6k · 2025-11-01

**Soundness:** 3
**Presentation:** 4
**Contribution:** 3
**Rating:** 6
**Confidence:** 5

**Summary:**

This paper presents ScaffAug, a scaffold-aware generative augmentation and reranking framework for ligand-based virtual screening. It addresses three key challenges—class imbalance, structural imbalance, and low scaffold diversity—by combining (1) scaffold-aware sampling and graph diffusion–based molecule generation, (2) self-training with pseudo-labels, and (3) MMR-based reranking to enhance structural diversity. Evaluations on five WelQrate datasets show consistent improvements over existing graph augmentation baselines across multiple screening metrics.

**Strengths:**

**Originality:** The work presents an original synthesis of scaffold-aware generative augmentation with MMR reranking, explicitly tackling class and structural imbalance in VS datasets.

**Quality:** Experiments are extensive and include ablations (e.g., Figure 4 on SAS vs UA sampling) and diversity analyses. The comparison against graph augmentation baselines is fair and supports the main claims.

**Clarity:** Writing is generally clear and coherent. The motivations for each module are explained in context, and figures effectively support the narrative.

**Significance:** The method offers a practical route for improving VS under class-imbalanced and scaffold-biased regimes, which is a real limitation in modern drug discovery pipelines.

**Weaknesses:**

**Originality:** Although combining augmentation, self-training, and reranking is novel in this context, each component is adapted from known ideas. A stronger theoretical justification for how these modules interact (e.g., information flow between SAS → GDM → self-training) would enhance originality.

**Quality:**

- No reported runtime efficiency or scalability results; this matters for large-scale screening campaigns.

- Lack of data leakage analysis during augmentation (i.e., ensuring augmented molecules from test scaffolds are excluded).

- DiGress is trained on 450 K unlabeled molecules; explaining why this subset (and not larger resources like ZINC or Enamine 100 M) was chosen would clarify generalizability.

**Clarity:** Some algorithmic details (e.g., Eq. 1 and Algorithm 3 notation) could be refined to avoid ambiguity.
Minor formula correction in Eq. 1 and  Algorithm 3 are self-referential and should likely be:

$G^t = m \odot s + (1 - m) \odot G^{t-1}$

**Significance:** Since protein structure information is ignored, the model may struggle for targets where ligand-only context is insufficient — a limitation already acknowledged by the authors in Section 5.

**Adjustments:** Convert "Table 3" to a "Figure 3".

**Suggestions:**

- Include before/after visualization of scaffold distribution to show how augmentation affects timesplit consistency.
- Evaluate runtime and scaling with dataset size.

**Questions:**

1- Could you provide runtime and computational resource comparisons (e.g., augmentation time per 1 K molecules)?
2- Why was the 450 K unlabeled molecule set chosen over larger public libraries (e.g., ZINC 15, Enamine 100 M as benchmarked in **[1]**)?

**References:**

**[1]** Graff et al., Accelerating high-throughput virtual screening through molecular pool-based active learning, 2021

---

> ### Author Response · Authors · 2025-11-22
> **Response to Weakness 1**
>
> # Weakness
>
> > 1. Although combining augmentation, self-training, and reranking is novel in this context, each component is adapted from known ideas. A stronger theoretical justification for how these modules interact (e.g., information flow between SAS $\rightarrow$ GDM $\rightarrow$ self-training) would enhance originality.
>
> Thank you for the insightful comments. We have discussed the motivation and design insights in the paper. ScaffAug begins by identifying three major challenges: (i) extreme class imbalance, (ii) structural imbalance among active scaffolds, and (iii) the need to retrieve diverse active compounds for virtual screening tasks. The scaffold-aware sampling and scaffold-extension modules are designed to address (i) and (ii): they oversample underrepresented active scaffolds and expand them using a scaffold-conditioned diffusion model, generating new molecules that strengthen the model’s ability to recognise true actives in the top-ranked list. The self-training module is intentionally simple, as it isolates the effect of the augmented data and demonstrates that the generated molecules are informative rather than noisy. Finally, the MMR reranking module addresses (iii) by explicitly balancing predicted activity and scaffold diversity in the final top-$k$ molecules.
>
> We also conduct thorough ablation studies that show how the interaction between ScaffAug's modules leads to state-of-the-art results. We include the ablation study for the three modules: augmentation module, self-training module, and reranking module, and we can discuss it more clearly in the main text.
>
> - **Augmentation module**. **Figure 4 and Figure 5** show that scaffold-aware sampling (SAS) directly targets structural imbalance by selectively expanding coverage around underrepresented active scaffolds in chemical space, rather than duplicating dense regions. Across four ranking metrics and three GNN backbones, ScaffAug ${ }_{\text {sas }}$ achieves the largest and most consistent gains over the oversampling baseline in Figure 4, which indicates that SAS is the key driver for improving enrichment when used as input to the downstream modules."
>
> - **Self-training module**. We evaluate a direct augmentation variant, ScaffAug ${ }_{d a}$, that assigns the active label to all generated molecules and merges them with the original dataset, and find that this strategy performs worse than naïve GCN training. In contrast, our self-training module, which filters and reweights generated molecules based on prediction confidence, consistently outperforms the GCN baseline across all five datasets and both split types (**Figure 6 and Table 3** in Appendix A.7), showing that the way augmented data flows from SAS/GDM into the classifier is crucial.
>
> - **Reranking module**. **Table 1** does not use reranking, so the gains there come purely from augmentation plus self-training. In **Figure 3**, we vary the trade-off parameter $\lambda$ for ScaffAug $_{\text {DeepGIN }}: \lambda=1.0$ corresponds to no reranking, while $\lambda<1.0$ activates MMR and puts more weight on scaffold diversity; as $\lambda$ decreases, SD100 improves consistently and EF100 stays similar or even improves in several assays. This shows that reranking further enhances scaffold diversity on top of the base ScaffAug model without harming early enrichment, confirming that each module contributes in a complementary way.

---

> ### Author Response · Authors · 2025-11-22
> **Response to Weakness 2, 3, 4, 5**
>
> > 2. No reported runtime efficiency or scalability results; this matters for large-scale screening campaigns.
>
> Thank you for the suggestion. Training DiGress on the full 450k unlabeled dataset takes approximately 1,450 seconds per epoch on an NVIDIA A100-SXM4-80GB. A Complete training of 300 epochs would take around 5 days, if we do not encounter any shared cluster resource constraints.
>
> We report the runtime efficiency of the augmentation module, specifically the scaffold-conditioned generation, across the WelQrate datasets, which range from roughly 60k to 300k molecules. Note that we sample $N$ scaffolds for scaffold extension, where $N$ euqls to 10% of the training size. This experiment is run on a single NVIDIA H100 80GB HBM3.
>
> | Name      | Num Generated Molecules | Runtime (s) |
> | --------- | ------------- | ----------- |
> | AID1798   | 3,642         | 258.14      |
> | AID463087 | 5,739         | 375.98      |
> | AID485290 | 16,868        | 1,087.11    |
> | AID488997 | 17,314        | 1,189.52    |
> | AID2689   | 18,268        | 1,152.63    |
>
> We omitted the runtime efficiency analysis for the self-training module because the computational overhead is marginal. As detailed in Appendix A.6, the additional cost of confidence-based pseudo-labelling stems from the iterative loop: (i) periodic forward passes on the unlabeled dataset, (ii) confidence filtering, and (iii) training on the expanded dataset. Since the time complexity of these steps scales linearly with the augmentation size, which we limit to ~10% of the labeled dataset, the total runtime remains comparable to naive training of prediction models.
>
> ---
>
> > 3. Lack of data leakage analysis during augmentation (i.e., ensuring augmented molecules from test scaffolds are excluded)
>
> Thank you for raising this important point. As clarified in Section 3.1 *Augmentation Module: Mitigating Class and Scaffold Imbalances*, we extract scaffolds **exclusively from active molecules in the training set** and use only these for scaffold extension during augmentation. While scaffolds from the training set may reappear in the test set under the **random split**, the **scaffold split** enforces that scaffolds in the training set do not appear in the validation or test sets, preventing leakage. The details of the data splits can be found in *Appendix A.3*.
>
> > 4. DiGress is trained on 450 K unlabeled molecules, explaining why this subset (and not larger resources like ZINC or Enamine 100 M ) was chosen would clarify generalizability.
>
> We acknowledge that ZINC15 and Enamine 100M are substantially larger datasets that cover many more compounds for virtual screening. However, training DiGress is computationally expensive. Training DiGress on 450k molecules for 300 epochs on a single A100 requires more than 5 days. Due to limited computing resources, we were unable to train DiGress on substantially larger datasets. We also preserve the diversity of these 450k molecules to cover a broad chemical space for training Digress, so it can generate structurally diverse and chemically valid compounds for our virtual screening task.
>
> Besides, ScaffAug's augmentation module we propose is agnostic to the specific molecular generative model. This means it can be directly paired with generative models pre-trained on large unlabeled datasets and still perform scaffold-extension tasks. SAFE [1] is a good example.
>
> [1] Noutahi, Emmanuel, Cristian Gabellini, Michael Craig, Jonathan S. C. Lim, and Prudencio Tossou. “Gotta Be SAFE: A New Framework for Molecular Design.” arXiv:2310.10773. Preprint, arXiv, December 10, 2023. https://doi.org/10.48550/arXiv.2310.10773.
>
> ---
> > 5. Some algorithmic details (e.g., Eq. 1 and Algorithm 3 notation) could be refined to avoid ambiguity. Minor formula correction in Eq. 1 and Algorithm 3 are self-referential and should likely be:
> > $$
> > G^t=m \odot s+(1-m) \odot G^{t-1}
>  $$
>
> Thank you for the suggestion. However, we believe that Eq. 1 and Alg. 3 convey the intended meaning. Algorithm 1 describes how we perform scaffold extension using DiGress, a graph diffusion model. At each reverse diffusion step (t), we first fix the scaffold by applying the mask (m), i.e., $G^{t} = m \odot s + (1 - m) \odot G^{t}$. We then sample the unknown regions of the molecular graph using the denoising network $p_{\theta}$, following $G^{t-1} \sim p_{\theta}(G^{t-1} \mid G^{t})$. This procedure ensures that the scaffold subgraph remains unchanged while the generative model fills in the remaining structure.

---

> ### Author Response · Authors · 2025-11-22
> **Response to Weakness 6, 7, 8, 9**
>
> > 6. Since protein structure information is ignored, the model may struggle for targets where ligand-only context is insufficient - a limitation already acknowledged by the authors in Section 5.
>
> We agree that incorporating high-quality protein structural information can substantially benefit models when such data are available. However, structure-based approaches also come with important limitations and uncertainties.
>
> - Not all drug targets are proteins, and many non-protein targets—such as RNA—lack accurate or widely validated structure-prediction tools. [2]
> - Current structure-based prediction tools (e.g., AlphaFold) tend to perform poorly when little or no multiple-sequence-alignment (MSA) information is available. This is a common scenario for understudied or de novo proteins. [3]
> - The global docking in the structure-based approaches is computationally expensive, limiting scalability for high-throughput applications.
>
> Protein structures typically reveal only the primary, well-studied binding site. **Ligand-based virtual screening(LBVS), however, is not constrained by knowledge of a specific binding site**. It can implicitly capture diverse binding modes, and it can more readily identify molecules with multi-target potentials. For example, Stokes et al. [4] demonstrated that a deep-learning model trained solely on ligand structures and bioactivity data could screen more than 107 million molecules and discover new antibacterial compounds that were structurally unlike known antibiotics, illustrating how ligand-based screening can uncover novel scaffolds and mechanisms of action without relying on predefined pocket information.
>
> [2] Yu AM, Choi YH, Tu MJ. RNA Drugs and RNA Targets for Small Molecules: Principles, Progress, and Challenges. Pharmacol Rev. 2020 Oct;72(4):862-898. doi: 10.1124/pr.120.019554. PMID: 32929000; PMCID: PMC7495341.
> [3] Aubel M, Eicholt L, Bornberg-Bauer E. Assessing structure and disorder prediction tools for de novo emerged proteins in the age of machine learning. F1000Res. 2023 Mar 29;12:347. doi: 10.12688/f1000research.130443.1. PMID: 37113259; PMCID: PMC10126731.
> [4] Stokes, Jonathan M., et al. “A Deep Learning Approach to Antibiotic Discovery.” Cell 180, no. 4 (2020): 688–702.e13. https://doi.org/10.1016/j.cell.2020.01.021.
>
> ---
>
> > 7. Include before/after visualization of scaffold distribution to show how augmentation affects timesplit consistency.
>
> Thank you for the helpful suggestion. In **Figure 5**, we present UMAP visualizations showing the distribution of active molecules before augmentation and the distribution of generated molecules after augmentation. This comparison shows how the augmentation module augments the underrepresented scaffolds. However, we are not entirely sure what the reviewer means by “timesplit consistency.” We would appreciate clarification so that we can address this point more precisely.
>
> ---
> > 8. Adjustments: Convert "Table 3" to a "Figure 3".
>
> Thank you for pointing this out. We have corrected the typo in the revised manuscript.
>
> ---
>
> > 9. Evaluate runtime and scaling with dataset size.
>
> Thank you for the comment. We have included the run-time analysis in the response to **Weakness 2** above.

---

> ### Author Response · Authors · 2025-11-22
> **Response to Question 1,2**
>
> # Questions
>
> > 1. Could you provide runtime and computational resource comparisons (e.g., augmentation time per 1 K molecules)?
>
> Thank you for the question. We have included the run-time analysis in the response to **Weakness 2** above.
>
>
> > 2. Why was the 450 K unlabeled molecule set chosen over larger public libraries (e.g., ZINC 15, Enamine 100 M as benchmarked in [1])?
>
> Thank you for the question. We have included the reasons for curating a 450K unlabeled dataset in the response to **Weakness 4**.

---

### Official Review · Reviewer_JNBY · 2025-11-01

**Soundness:** 2
**Presentation:** 3
**Contribution:** 3
**Rating:** 4
**Confidence:** 4

**Summary:**

The paper proposes an augmentation method designed to enhance virtual screening results. The method addresses issues with training data imbalance between active and inactive compounds, the overrepresentation of certain scaffolds, and the diversity of the identified hits. To augment the training dataset with more positive data, scaffolds from active compounds are extracted and sampled with replacement, using probabilities inversely proportional to their frequency in the active compounds. Next, additional examples are generated using the DiGress model, which retains the scaffold structure. To train the activity prediction model, pseudo-labeling is conducted after a warm-up phase to increase the training dataset. The ranking of molecules is performed using an MMR-based approach, which, in addition to predicted activity, considers the similarity to molecules that have already been selected. The proposed approach achieves strong results on a subset of the WelQrate dataset.

**Strengths:**

- The paper is written in a clear way, and the motivation is explained well.
- The problem this work addresses is important, as making virtual screening more efficient would help in finding new therapeutic molecules faster.
- The experiments are conducted across five datasets and two types of data splitting methods, repeated three times to report standard deviation values.
- ScaffAug demonstrates strong performance across all the selected datasets, showing promise in the virtual screening setup.
- Augmenting scaffolds is a novel strategy addressing class imbalance in biological activity prediction.
- The impact of MMR reranking is shown in Figure 3. Although this reranking strategy results in lower enrichment factor values, the diversity of the selection is significantly improved.

**Weaknesses:**

- Generating molecules with the same scaffolds does not help with the scaffold diversity of discovered molecules. Actually, it might bias the model to predict molecules with the same scaffolds more often than new active compounds with novel scaffolds. This can be a big disadvantage for drug discovery campaigns wanting to avoid patent-protected chemical spaces.
- The indexing in line 16 of Algorithm 1 can be misleading. Now, this notation suggests that the algorithm returns the first $N$ scaffolds from the input.
- The chemical validity or synthesizability of the molecules generated with DiGress is not reported in the paper. Providing some examples of the generated molecules would also be helpful.
- The experiments lack simple baselines where molecules are ranked by $f_\phi$ predictions with and without training on pseudo-labeled examples and with and without augmentation.
- (minor) The problem definition involving binary classification does not correspond well to the hit selection by ranking problem. Perhaps VS should be defined rather as a ranking problem or finding $k$ molecules with the highest probability of being active, which can be achieved by using a classification model.
- (minor) In line 179, there seems to be part of the text missing.

**Questions:**

1. Should the sigmoid function be applied in line 7 of Algorithm 2 if $p_i$ is already a probability (presumably since $f_\phi$ is a binary classification model)?
2. How do you approach scaffolds that occur both in active and inactive compounds? Do you still use them to generate augmented samples?

---

> ### Author Response · Authors · 2025-11-22
> **Response to Weaknesses 1, 2, 3**
>
> # Weakness
> > 1. Generating molecules with the same scaffolds does not help with the **scaffold diversity** of discovered molecules. Actually, it might bias the model to predict molecules with the same scaffolds more often than new active compounds with novel scaffolds. This can be a big disadvantage for drug discovery campaigns wanting to avoid patent-protected chemical spaces.
>
> Thank you for the insightful comments. We fully acknowledge that identifying active, novel molecules remains the major challenge in any virtual screening task. Accordingly, medicinal chemists respond to this challenge by emphasizing scaffold diversity in their curated datasets before screening. That is, they select compounds to cover as many distinct scaffolds as possible, thereby improving the chances of discovering novel active molecules[1]. For example, the Vanderbilt Institute of Chemical Biology V-HTS facility explicitly describes its Diverse Collections as “broad-coverage, chemically diverse sets meant to represent as much chemical space as possible without targeting a specific protein family or therapeutic area.”
>
> When analyzing the WelQrate datasets, which are curated from real bioassays, we observed substantial structural imbalance: some scaffolds occur far more frequently among active molecules than others, as shown in Figure 4. To address this issue, we apply scaffold-aware sampling to strengthen the representation of underrepresented scaffolds within the active class. Through experiments, we demonstrate that this augmentation strategy reduces the model’s bias to overfit to a small number of dominant scaffolds and increases the overall scaffold diversity of active molecules in the training set. Scaffold hopping can be further explored to create a more diverse augmentation dataset in future work. Scaffold hopping[3], which generates novel chemotypes by replacing the core molecular scaffold while preserving biological activity, can be explored in future work to further enrich the diversity of the augmented dataset.
>
> We also carefully examined the molecules generated during augmentation and found that DiGress can generate novel scaffolds not present in the training set, even when the sampled scaffolds are fixed in the reverse process. For example, in the AID1798 dataset, the training set contains 93 unique scaffolds, whereas the augmentation process yields 543 additional scaffolds. Appendix A.8 provides an example of a sampled active scaffold along with a ranked list of generated scaffolds absent from the training set, ordered by similarity scores. These observations indicate that the generative model effectively explores the chemical space surrounding underrepresented scaffolds.
>
> [1] Krier M., Bret G., Rognan D. Assessing the Scaffold Diversity of Screening Libraries. J. Chem. Inf. Model. 2006;46(2):512-524.
> [2] Vanderbilt Institute of Chemical Biology – High-Throughput Screening Core. Screening | V-HTS Compound Collections – Diverse Collections. Accessed Nov 18 2025. https://medschool.vanderbilt.edu/vicb/hts/services-copy/
> [3] Brown, Nathan. (2013). Scaffold Hopping in Medicinal Chemistry. 10.1002/9783527665143.ch01.
>
> ---
>
> > 2. The indexing in line 16 of Algorithm 1 can be misleading. Now, this notation suggests that the algorithm returns the first $N$ scaffolds from the input.
>
> Thank you for the suggestion. We have fixed this ambiguity of Algorithm 1 in the revised manuscript.
>
> ---
>
> > 3. The **chemical validity or synthesizability** of the molecules generated with DiGress is not reported in the paper. Providing some examples of the generated molecules would also be helpful.
>
> Thank you for raising this comment. The validity rate of our trained DiGress model is 0.85, with a uniqueness of 1.0 when sampling 10,000 molecules. Scaffold extension is inherently more challenging, which leads to a lower validity rate. Below, we report the mean validity rate across splits for all datasets. Since the reduced validity may affect the effectiveness of scaffold-aware sampling, we iteratively generate molecules for each sampled scaffold until a valid molecule is obtained. **Only chemically valid molecules are included in our augmentation dataset.** We have also provided some examples of generated molecules in *Appendix A.8*.
>
> | Dataset   | Validity Rate|
> | --------- | --------------------- |
> | AID1798   | 0.6244                |
> | AID2689   | 0.5314                |
> | AID463087 | 0.6117                |
> | AID485290 | 0.6953                |
> | AID488997 | 0.6251                |

---

> ### Author Response · Authors · 2025-11-22
> **Response to Weakness 4, 5, 6**
>
> > 4. The experiments lack simple baselines where molecules are ranked by $f_\phi$ predictions with and without training on pseudo-labeled examples and **with and without augmentation**.
>
> Thank you for pointing this out. Although we do not explicitly present the results of training without pseudo-labeled examples in the main text, this setting is included as the oversampling baseline in Figure 4. This baseline corresponds to training the predictive model solely on the original labeled data, without any augmentation or self-training. Specifically, Figure 4 reports the mean percentage improvements of our two augmentation variants over this oversampling baseline across four ranking metrics. For each backbone model (GCN, GAT, and DeepGIN), the baseline is represented by the zero reference line, while the colored bars show the relative performance gains achieved by our augmentation strategies. As illustrated in the figure, $\text{ScaffAug}_\text{sas}$ consistently shows the best improvements over the oversampling baseline across nearly all metrics and architectures. We will make this comparison clearer in the revised manuscript.
>
> ---
>
> > 5. (minor) The problem definition involving binary classification does not correspond well to the hit selection by ranking problem. Perhaps VS should be defined rather as a ranking problem or finding $k$ molecules with the highest probability of being active, which can be achieved by using a classification model.
>
> Thank you for the insightful comment. We initially describe the task as a binary classification problem because we train the predictive model using a binary cross-entropy loss. However, rather than evaluating the model on its ability to predict discrete labels in the test set, we rank its predicted scores and assess the quality of the top-ranked molecules. This better reflects real-world drug discovery workflows, where chemists primarily focus on and synthesize the highest-scoring candidates, such as those with the most favorable binding affinity predicted by the docking tools, for further experiments.
>
> > 6. (minor) In line 179, there seems to be part of the text missing.
>
> Thank you for pointing this out. We have corrected it in the revised manuscript.

---

> > ### Author Response · Authors · 2025-11-22
> > **Response to Questions 1, 2**
> >
> > # Questions
> > > 1. Should the sigmoid function be applied in line 7 of Algorithm 2 if $p_i$ is already a probability (presumably since $f_\phi$ is a binary classification model)?
> >
> > We appreciate this helpful question. We evaluate virtual screening as a **ranking** problem, and therefore we do not apply the sigmoid function when computing the four ranking metrics: logAUC, BEDROC, EF, and DCG.  In Algorithm 2, the sigmoid is used solely within the MMR-based ranking procedure to map prediction scores into the $(0, 1)$ range, ensuring they are on the same scale as the similarity scores.
> >
> > > 2. How do you approach scaffolds that occur both in active and inactive compounds? Do you still use them to generate augmented samples?
> >
> > If a scaffold appears in both active and inactive compounds, we still include it in the augmentation process as long as it is present in the active distribution. Once a scaffold is associated with active molecules, we assume it contributes to the binding signal against the target and carries desirable molecular properties such as solubility, membrane permeability, stability, and metabolic profile. Therefore, it remains valuable to augment scaffolds that occur in both active and inactive compounds.

---

### Official Review · Reviewer_28QU · 2025-11-02

**Soundness:** 3
**Presentation:** 3
**Contribution:** 3
**Rating:** 6
**Confidence:** 3

**Summary:**

This work presents ScaffAug: a framework for improving ligand-based virtual screening by utilizing synthetic data for pre-training. This data comes from completion of scaffolds occurring in highly active molecules. Through quantitative experiments, authors show this approach - pre-training followed by inference-time post-processing - is effective in achieving a diverse coverage of active compounds.

**Strengths:**

**(S1)**: Making virtual screening more robust to data and scaffold imbalance is a worthwhile pursuit, and the paper makes a good attempt at resolving these challenges. From an ML point of view the method design looks sound.

**(S2)**: The paper is reasonably written and mostly easy to understand.

**Weaknesses:**

**(W1)**: There are a few parts of the work that would benefit from further clarification:

- **(W1a)**: I am wondering what is the rationale behind the reranking improving results. While this method would be bound to improve e.g. the internal diversity of the returned set, it is not clear to me that improved diversity would necessarily lead to better results. It does seem to be the case given Figure 3, and there is a clear sweet-spot in the middle of reranking strength $\lambda$, which I found surprising; I expected increasing diversity will necessarily lead to deteriorating quantitative performance.

- **(W1b)**: In the text, authors say that in their augmentation method they "preserve the scaffolds of active compounds and generate new molecules with a graph diffusion model". However, carefully reading Algorithm 1 reveals the process is more involved, as instead clusters are scored based on the proportion of actives within, and (as far as I understand) a scaffold coming from an inactive molecule can also be selected as long as it got clustered together with a lot of scaffolds coming from actives.

---

**Other comments**

**(O1)**: It would be good to also widely refer and relate to scaffold-based and scaffold-constrained generative models [1, 2].

**Nitpicks**

- Typos in "bsaeline" (caption of Figure 1), "generate synthesis data" (Section 3.1).

- In Section 3.1, last sentence of the first paragraph appears to be broken.

- In Section 4.1, "we utilize a subset of the comprehensive WelQrate(Wang et al., 2024) dataset is a high-quality dataset of 9 bioassays" seems broken (also missing space before citation).

- Very minor, but in my view "generative AI" is mostly a marketing term nowadays, it would be more typical to say "generative models".

**References**

[1] "Learning to Extend Molecular Scaffolds with Structural Motifs"

[2] "Scaffold-based molecular design using graph generative model"

**Questions:**

See the "Weaknesses" section above for specific questions.

---

> ### Author Response · Authors · 2025-11-22
> **Response to Weaknesses**
>
> > (W1a): I am wondering what is the **rationale behind the reranking improving results**. While this method would be bound to improve e.g., the internal diversity of the returned set, it is not clear to me that improved diversity would necessarily lead to better results. It does seem to be the case given Figure 3, and there is a clear sweet-spot in the middle of reranking strength, which I found surprising; I expected increasing diversity will necessarily lead to deteriorating quantitative performance.
>
> We greatly appreciate the insightful comment. Yes, we agree that increasing diversity in the top predictions generally leads to a decrease in quantitative performance. As shown in Figure 3, the $EF_{100}$ score decreases when we reduce $\lambda$ to increase scaffold diversity among the top 500 predicted molecules. The following reasons can explain the “sweet spot” in the middle of the reranking-strength range:
>
> * **Scaffold split:** In the scaffold split, we partition molecules into train/validation/test sets by grouping compounds that share the same core scaffold, and large scaffold groups are typically assigned to the training set. As a result, ScaffAug is evaluated on structurally novel molecules, and the test set is more diverse because it contains compounds less likely to share scaffolds with the training data. In this context, the model needs to explore broader chemical space to identify active molecules. Empirically, we find that increasing scaffold diversity does not sharply degrade performance and can even improve EF scores, especially for the AID2689 dataset.
>
> * **Random split:** It is indeed surprising to observe a similar trend under the random split. We believe this is because the active molecules in the test set are still diverse, as they represent only a small fraction of the overall active pool. Under this condition, the intuition that increasing scaffold diversity helps discover novel active molecules is reflected in the empirical results.
>
> ---
>
> > (W1b): In the text, authors say that in their augmentation method they "preserve the scaffolds of active compounds and generate new molecules with a graph diffusion model". However, carefully reading Algorithm 1 reveals the process is more involved, as instead clusters are scored based on the proportion of actives within, and (as far as I understand) a scaffold coming from an inactive molecule can also be selected as long as it got clustered together with a lot of scaffolds coming from actives.
>
> Thank you for pointing out the confusion caused by Algorithm 1. In our actual implementation, we only extract **scaffolds from active molecules** in the training set and encode them using ECFP fingerprints. We then apply K-means clustering to assign scaffold labels, with the optimal number of clusters selected based on silhouette scores. Importantly, scaffolds originating from inactive molecules are not used in this process. We have corrected Algorithm 1 in the revised manuscript to accurately reflect this procedure.
>
> ---
>
> > (O1)It would be good to also widely refer and relate to scaffold-based and scaffold-constrained generative models [1, 2].
> > [1] "Learning to Extend Molecular Scaffolds with Structural Motifs"
> > [2] "Scaffold-based molecular design using graph generative model"
>
> Thank you for the suggestion. We agree that these two works are directly relevant to scaffold-conditioned generation. Accordingly, we will expand Section 2.2 (Related Work) to include a discussion of these papers.
>
> [1] Lim, Jaechang, Sang-Yeon Hwang, Seokhyun Moon, Seungsu Kim, and Woo Youn Kim. 2020. “Scaffold-Based Molecular Design with a Graph Generative Model.” Chemical Science 11 (4): 1153–64. https://doi.org/10.1039/C9SC04503A.
>
> [2] Krzysztof Maziarz, Henry Jackson-Flux, Pashmina Cameron, Finton Sirockin, Nadine Schneider, Nikolaus Stiefl, Marwin Segler & Marc Brockschmidt. Learning to Extend Molecular Scaffolds with Structural Motifs. arXiv:2103.03864 (ICLR 2022).
>
> ---
>
> > Nitpicks
> > Typos in "bsaeline" (caption of Figure 1), "generate synthesis data" (Section 3.1).
> > In Section 3.1, last sentence of the first paragraph appears to be broken.
> > In Section 4.1, "we utilize a subset of the comprehensive WelQrate (Wang et al., 2024) dataset is a high-quality dataset of 9 bioassays" seems broken (also missing space before citation).
> > Very minor, but in my view "generative AI" is mostly a marketing term nowadays, it would be more typical to say "generative models".
>
> Thank you for the careful reading and for pointing out these issues. We have corrected all noted typos, fixed the broken sentences in Sections 3.1 and 4, and replaced “generative AI” with “generative models.” These changes are included in the revised manuscript.

---

### Author Response · Authors · 2025-12-03
**ScaffAug's Rebuttal Summary**

We sincerely thank the area chair for their thoughtful evaluation of our submission at this very moment. We also appreciate the reviewers for their detailed and constructive feedback, which has been invaluable in improving the clarity and completeness of our work. In this general comment, we will summarize the common strengths raised by reviewers and how we address the reviewers' concerns

# Strengths
- Clear motivation and well-structured presentation
- Originality of the ScaffAug framework
- Comprehensive Empirical Evaluations

# Rebuttals
## 1. Novelty of ScaffAug
> Reviewer zpNN questioned whether ScaffAug provides sufficient methodological innovation beyond combining existing components.

- We argue that ScaffAug provides method-level contributions tailored specifically to the challenges of real-world ligand-based virtual screening(VS).

- ScaffAug is built from a *data-centric analysis* of realistic HTS datasets and directly addresses three major domain challenges:
**(i) extreme class imbalance, (ii) structural imbalance among active scaffolds, and (iii) the need for identifying diverse active molecules.** Scaffold-aware sampling and scaffold extension target (i) and (ii) by oversampling underrepresented active scaffolds and expanding them with a scaffold-conditioned diffusion model, generating new molecules that meaningfully improve the model’s ability to identify active molecules in the top list. The self-training strategy isolates and demonstrates the quality of the augmentation dataset. The MMR reranking module addresses (iii) by explicitly trading off prediction scores and scaffold diversity in the predictive top-k set.

## 2. Compare ScaffAug with InstructMol [1]
> Reviewer zpNN asked us to compare ScaffAug with InstructMol.

We acknowledge that InstructMol is a novel semi-supervised method that leverages unlabeled data via an instructor–student mechanism. However, InstructMol has important limitations when applied to real-world VS:
- **Poor generalization for VS datasets**: In VS datasets where the active rate is <1% and structural imbalance presents, InstructMol underperforms ScaffAug. Using the same DeepGIN backbone and ZINC15_250K unlabeled pool with default hyperparameters, ScaffAug **consistently outperforms InstructMol** on all but the AID488997 scaffold split. The results are included in the new manuscript.
- **Inefficient use of large unlabeled datasets**: InstructMol assigns pseudo-labels to large unlabeled sets (250k–1M) and trains on them despite most unlabeled molecules being irrelevant to the task. This leads to computational inefficiency and poor scalability to settings like WelQrate, where the labeled set reaches ~300k samples. **In contrast, ScaffAug generates small but highly targeted unlabeled data** through scaffold-aware sampling and scaffold extension, directly addressing data-centric challenges. This highlights the scalability of ScaffAug.

Together, these factors highlight that ScaffAug provides methodological innovation tailored for realistic VS scenarios and achieves substantially better performance and scalability than InstructMol.

## 2. Ablation of the Three Modules
> Several reviewers (zpNN, BG6k, and JNBY) requested explicit ablations for the augmentation, self-training, and reranking modules.

**We clarify that the paper already includes all three ablations and summarize the key findings below. We will improve the narrative of ablation studies in the revised manuscript.**

## 3. Other Minor Weaknesses and Questions

### (1) Chemistry-related questions.

We explain the following questions, which are essential for our paper.
- Why is scaffold diversity critical for virtual screening success(raised by zpNN)?
- Why is ligand-based virtual screening essential and effective in realistic drug discovery(raised by BG6k)?

### (2) Time-Cost and Scalability Analysis
- **Pretraining DiGress.** Training DiGress on the 450k unlabeled set takes ~1,450 s/epoch on an A100‑80GB; 300 epochs require about 5 days. This is a one‑time offline cost that can be reused across targets.
- **Per‑dataset augmentation cost.** Scaffold‑conditioned generation on a single H100‑80GB takes: 258–376 s to generate 3.6–5.7k molecules, and 1,087–1,190 s to generate 16.9–18.3k molecules across the five WelQrate assays.

This allows efficient augmentation in ScaffAug if we have a solid generative model.

### (3) Validity rate and quality of generated molecules
We report that the trained DiGress model achieves 0.85 validity and 1.00 uniqueness when sampling 10k molecules unconditionally. We acknowledge that the scaffold extension is a harder task, and the mean validity rate across splits is decreasing. **Therefore, for each sampled scaffold, we resample until at least one valid molecule is obtained and ensure that **only valid molecules** enter the augmentation set**.

### (4) Typos and Writing issues
We have addressed these issues in the updated manuscript.

---

### Note · Program_Chairs · 2026-01-17
**Submission Desk Rejected by Program Chairs**

The following references in this submission do not refer to real documents and/or have major errors in bibliographic information:

 Ricardo P Nobrega, Minkai Wu, Cho-Jui Hsieh, and Eric P Xing. Midi: Mixed diffusion for structure-based drug design. In Advances in Neural Information Processing Systems, volume 35, pp. 22198-22211, 2022.